# Self-Supervision Enhanced Feature Selection with Correlated Gates

**Changhee Lee**[*]
Chung-Ang University, Korea
changheelee@cau.ac.kr

**Fergus Imrie**[*]
UCLA, USA
imrie@ucla.edu

**Mihaela van der Schaar**
University of Cambridge, UK
UCLA, USA
Alan Turing Institute, UK
mv472@cam.ac.uk

## Abstract

Discovering relevant input features for predicting a target variable is a key scientific question. However, in many domains, such as medicine and biology, feature selection is confounded by a scarcity of labeled samples coupled with significant correlations among features. In this paper, we propose a novel deep learning approach to feature selection that addresses both challenges simultaneously. First, we pre-train the network using unlabeled samples within a self-supervised learning framework by solving pretext tasks that require the network to learn informative representations from partial feature sets. Then, we fine-tune the pre-trained network to discover relevant features using labeled samples. During both training phases, we explicitly account for the correlation structure of the input features by generating correlated gate vectors from a multivariate Bernoulli distribution. Experiments on multiple real-world datasets including clinical and omics demonstrate that our model discovers relevant features that provide superior prediction performance compared to the state-of-the-art benchmarks in practical scenarios where there is often limited labeled data and high correlations among features.

## 1 Introduction

High-dimensional datasets are increasingly prevalent in a variety of fields, including critical domains such as medicine and biology. Discovering features responsible for the target variable is an essential but challenging step in the analysis of such data. For example, while next generation sequencing can detect the expression of tens of thousands of genes per sample (e.g., RNA-seq), many genetic disorders stem from the variation in only a few groups of related genes (Jackson et al., 2018). Identification of such disease-related factors is crucial for the design of therapeutic treatments. Furthermore, feature selection has additional benefits, including reduced experimental costs, greater interpretability, and improved model generalization (Min et al., 2014; Ribeiro et al., 2016).

However, the high-dimensional and often noisy nature of such data prevents relevant features from being readily discovered. Moreover, in many domains there is a scarcity of label information due to cost, privacy reasons, or experimental study design. While deep learning approaches have resulted in improvements in feature selection (e.g., Yamada et al. (2020); Lemhadri et al. (2021), see Related Work), such methods typically assume access to many labeled samples. With low sample size, feature selection methods, in particular those employing deep learning, have shown a propensity to overfit high-dimensional data (Kuncheva et al., 2020).

This issue is further exacerbated by the inherent structure of such data. Many (high-dimensional) datasets exhibit substantial inter-correlations or multicollinearity – i.e., there exist features that are (highly) correlated among themselves – which impacts the performance of feature selection methods (Chong & Jun, 2005; Katrutsa & Strijov, 2017; Belsley et al., 2005). In particular, this structure can cause parameter estimates to be unstable (Dobson & Barnett, 2018) and, in the extreme, can prevent variable effects from being separated, confounding the problem of feature selection (Meloun et al., 2002). Existing deep learning-based methods for feature selection do not (explicitly) consider the correlations between features which will often result in the selection of redundant features.

---

[*]Equal contribution

When labeled samples are scarce, large numbers of unlabeled samples are often available (e.g., Perez-Riverol et al. (2019)). However, the importance of unlabeled data has been overlooked in feature selection despite its potential to prevent the model selection process from overfitting by allowing informative feature representations to be learned. Therefore, the development of methods for feature selection that can exploit both labeled and unlabeled data is of great practical importance.

**Contributions.** We propose a novel method for feature selection that addresses two key challenges in real-world scenarios: limited labeled data and correlated features. We use a self-supervised approach to train an encoder using unlabeled data via two pretext tasks: feature vector reconstruction and gate vector estimation. This pre-conditions the encoder to learn informative representations from partial feature sets, aligning the self-supervision with the model selection process of the downstream feature selection task. In addition, we introduce a novel gating procedure that accounts for the correlation structure of the input features. This ensures the pretext tasks remain challenging by preventing the model from memorizing trivial relations between features. Moreover, unlike previous deep learning-based feature selection methods, the correlated gate vectors encourage our method to select the most relevant features by making multiple correlated features compete against each other.

We validate our approach through experiments on synthetic and multiple real-world datasets, including clinical and omics, where only a small number of labeled samples are available. Our model discovers relevant features that provide superior prediction performance compared to state-of-the-art benchmarks, and we corroborate these features with supporting medical and scientific literature.

## 2 RELATED WORK

**Feature Selection Methods.** Feature selection is a well-studied problem, with a number of proposed solutions including wrapper (Kohavi & John, 1997) and filter (Liu & Setiono, 1996; Kira & Rendell, 1992) methods. Recent advances in deep learning provide an elegant way of training embedded feature selection methods by jointly learning to perform feature selection while training a prediction network (Huang et al., 2020). These methods learn to perform the non-differentiable process of selecting feature subsets by approximating it either with Lasso or elastic net penalization (Li et al., 2016), using an MCMC sampling approach (Liang et al., 2018), or more recently with continuous relaxation using independent Gaussian random variables (Yamada et al., 2020). However, supervised feature selection methods can fail to identify relevant features when limited labeled samples are available due to overfitting (Kuncheva et al., 2020), impacting their suitability in many real-world scenarios. Moreover, these methods do not consider the underlying correlation structure of the input features which can be problematic when selecting relevant features among (highly) correlated ones.

Relatively few feature selection methods use unlabeled samples. Abid et al. (2019) used autoencoders to identify a pre-specified number of features that are sufficient for reconstructing the data. Lindenbaum et al. (2020) improved the well-known Laplacian score (He et al., 2005) by selecting feature subsets that better capture the "local structure" of the data. However, both approaches are fully unsupervised and, without the guidance of label information, can fail to identify features relevant to the target outcome. While a number of semi-supervised feature selection methods have been proposed (Sheikhpour et al., 2017), they have typically been extensions of traditional methods, such as a Laplacian score with modified affinity scores using labels (Zhao et al., 2008) or manifold regularization based on linear SVMs (Dai et al., 2013). To the best of our knowledge, this is the first deep learning framework that fully utilizes both labeled and unlabeled samples for feature selection.

**Self-Supervised Learning.** Self-supervised learning methods create (weak) supervised signals from unlabeled data, employing contrastive learning or pretext task(s) to provide surrogate labels. While self-supervised learning has found success in computer vision (Chen et al., 2020) and natural language processing (Devlin et al., 2018), the tabular domain, which is most relevant in feature selection, has been largely neglected. Methods for tabular data seek to reconstruct data either based on a corrupted sample alone (Vincent et al., 2008; Arik & Pfister, 2019; Yin et al., 2020) or with knowledge of which entries have been corrupted (Pathak et al., 2016). Recently, Yoon et al. (2020) jointly learn to recover the original sample and predict the mask vector used to corrupt the data.

While our pretext tasks are also reconstruction-based, we propose a novel method for generating the gate vector used to produce the input feature vector. This is particularly important when there exists substantial correlation between features (see Section 5). All existing methods have used indepen-

dent, uniform sampling; however, this allows highly correlated features to be readily reconstructed since it is likely that not all such features will be corrupted. We prevent this by incorporating the correlation structure into the gating procedure. In this work, we employ pretext tasks that mirror the goal of the feature selection process: the input to the encoder is a subset of the features selected at random. Employing partial feature sets for self-supervised learning encourages the encoder to learn better representations of these partial feature sets. This in turn benefits learning both the model selection function and the feature selection step since they are also trained using partial features sets.

## 3 PROBLEM FORMULATION

Let $\mathbf{X} = (X_1, \cdots, X_p) \in \mathcal{X}^p$ and $Y \in \mathcal{Y}$ be random variables for the high-dimensional input features (e.g., gene expressions) and the target outcome (e.g., disease traits) whose realizations are denoted as $\mathbf{x} = (x_1, \cdots, x_p)$ and $y$, respectively. Throughout the paper, we will often use lower-case letters to denote realizations of random variables.

Embedded feature selection aims to select a subset, $\mathcal{S} \subset [p]$, of features that are relevant for predicting the target as part of the model selection process. Denote $*$ be any point not in $\mathcal{X}$ and define $\mathcal{X}_{\mathcal{S}} = (\mathcal{X} \cup \{*\})^p$.[1] Then, given $\mathbf{X} \in \mathcal{X}^p$, the selected subset of features can be denoted as $\mathbf{X}_{\mathcal{S}} \in \mathcal{X}_{\mathcal{S}}$ where $x_{\mathcal{S},k} = x_k$ if $k \in \mathcal{S}$ and $x_k = *$ if $k \notin \mathcal{S}$. Let $f : \mathcal{X}_{\mathcal{S}} \to \mathcal{Y}$ be a function in $\mathcal{F}$ that takes as input the subset $\mathbf{X}_{\mathcal{S}}$ and outputs $Y$. Then, selecting relevant features can be achieved by solving the following optimization problem:

$$\underset{f \in \mathcal{F}, \, \mathcal{S} \subset [p]}{\text{minimize}} \quad \mathbb{E}_{\mathbf{x}, y \sim p_{XY}} \left[ \ell_Y \big( y, f(\mathbf{x}_{\mathcal{S}}) \big) \right] \qquad \text{subject to} \quad |\mathcal{S}| \leq \delta \tag{1}$$

where $\delta$ constrains the number of selected features and $\ell_Y(y, y') = -\sum_{c=1}^{C} y_c \log y'_c$ for $C$-way classification tasks (i.e., $\mathcal{Y} = \{1, \cdots, C\}$) and $\ell_Y(y, y') = \|y - y'\|_2^2$ for regression tasks (i.e., $\mathcal{Y} = \mathbb{R}$).[2] Unfortunately, the combinatorial problem in (1) becomes intractable for high-dimensional data as the search space increases exponentially with $p$. Hence, we instead focus on a relaxation by converting the combinatorial search in (1) into a search over the space of possibly correlated binary random variables. It is worth highlighting that unlike existing work (e.g., Yamada et al. (2020); Yoon et al. (2019)), we do not assume independence among these random variables.

Let $\mathbf{M} = (M_1, \cdots, M_p) \in \{0, 1\}^p$ be binary random variables governed by distribution $p_M$, whose realization $\mathbf{m}$ is referred to as the *gate vector*, for indicating selection of the corresponding features. Then, the selected features given gate vector $\mathbf{m}$ can be written as

$$\tilde{\mathbf{x}} \triangleq \mathbf{m} \odot \mathbf{x} + (1 - \mathbf{m}) \odot \bar{\mathbf{x}} \tag{2}$$

where $\bar{\mathbf{x}} = \mathbb{E}_{\mathbf{x} \sim p_X}[\mathbf{x}]$ and $\odot$ indicates element-wise multiplication. Here, we replace not-selected features with their means to resolve any issue when a feature having zero value (e.g., turned-off genes) has specific meaning. Finally, we can approximately achieve (1) by jointly learning the model selection $f$ and the gate vector distribution $p_M$ based on the following optimization problem:

$$\underset{f, \, p_M}{\text{minimize}} \quad \mathbb{E}_{\mathbf{x}, y \sim p_{XY}} \mathbb{E}_{\mathbf{m} \sim p_M} \left[ \ell_Y \big( y, f(\tilde{\mathbf{x}}) \big) + \beta \|\mathbf{m}\|_0 \right] \tag{3}$$

where $f$ is implemented as a neural network and $\beta$ is a balancing coefficient that controls the number of features to be selected.

**Challenges.** In practice (especially in the healthcare domain), there are two main challenges that confound selecting relevant features for predicting the target: (i) inter-correlation or multicollinearity, which is the existence of features that are (highly) correlated among themselves, and (ii) the absence of sufficient labeled samples. These challenges make embedded feature selection vulnerable to overfitting. More specifically, model selection (i.e., learning $f$) and feature selection (i.e., learning $p_M$) are conducted jointly. Either one being overfit to correlated or noisy irrelevant features will end up discovering spurious relations and poor prediction. For instance, if the network is biased toward irrelevant features, the selection probability (i.e., importance) of those features will be increased as if those features were "discriminative" or "predictive", and vice versa.

---

[1]To enable neural networks to be trained with varying feature subsets, we set $* = \bar{\mathbf{x}}$.

[2]For $\mathcal{Y} = \{1, \cdots, C\}$, we will occasionally abuse notation and write $y_c$ to denote the $c$-th element of the one-hot encoding of $y$ when clear in the context.

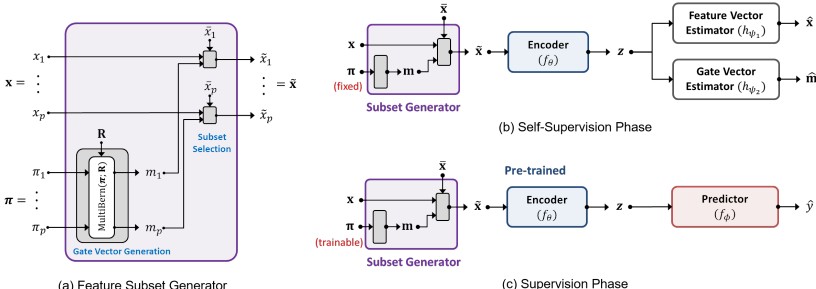

Figure 1: An illustration of SEFS. The selection probability in the Feature Subset Generator is fixed and treated as a hyper-parameter during Self-Supervision Phase while it is updated during Supervision Phase using a reparameterization trick.

## 4 METHOD: SELF-SUPERVISION ENHANCED FEATURE SELECTION

In this section, we describe our novel feature selection method, which we refer to as **S**elf-supervision **E**nhanced **F**eature **S**election (SEFS). To address the challenges described above, we propose a novel gate vector generation process modeled by a *multivariate Bernoulli* distribution where dependencies among gates are transferred from the correlation structure of the input features, and a two-step training procedure that utilizes both labeled and unlabeled samples.

Our approach is defined by the *selection probability* $\boldsymbol{\pi}$, that governs the gate vector generation process, and the model selection function $f$, that we decompose into an *encoder* $f_\theta$ and a *predictor* $f_\phi$, i.e., $f = f_\phi \circ f_\theta$, as illustrated in Figure 1. To utilize both labeled and unlabeled samples, these components are updated via a two-step training procedure:

- *Self-Supervision Phase*: the encoder $f_\theta$ is pre-trained to learn favorable representations for feature selection using unlabeled samples via solving two pretext tasks specifically designed to take only subsets of the features as input.

- *Supervision Phase*: the learned representations from the Self-Supervision Phase are adopted to solve the objective in (3) – i.e., updating the encoder $f_\theta$, the predictor $f_\phi$, and the selection probability $\boldsymbol{\pi}$ – for discovering task-relevant features using labeled samples.

### 4.1 MULTIVARIATE BERNOULLI GATE VECTORS USING GAUSSIAN COPULA

Instead of assuming independence in the gating mechanism (2), we model the gate vector distribution $p_M$ as a *multivariate Bernoulli* distribution where dependencies among the gates are determined by the correlation structure of the input features. To this goal, we use *Gaussian copula* (Nelsen, 2007) to construct a valid multivariate Bernoulli distribution. The Gaussian copula is a multivariate cumulative distribution function (CDF) of random variables $(U_1, \cdots, U_p)$ over the unit cube $[0, 1]^p$ with uniform marginals, where $U_k \sim \texttt{Uniform}(0, 1)$ for $k \in [p]$. Formally, given the correlation matrix $\mathbf{R} \in [-1, 1]^p$ that captures the correlation structure of $\mathbf{X}$, the Gaussian copula is defined as:

$$C_{\mathbf{R}}(U_1, \cdots, U_p) = \Phi_{\mathbf{R}}(\Phi^{-1}(U_1), \cdots, \Phi^{-1}(U_p)) \tag{4}$$

where $\Phi_{\mathbf{R}}$ stands for the joint CDF of a multivariate Gaussian distribution with mean zero vector and correlation matrix $\mathbf{R}$, and $\Phi^{-1}$ is the inverse CDF of the standard univariate Gaussian distribution.

Define $\boldsymbol{\pi} = (\pi_1, \cdots, \pi_p) \in [0, 1]^p$ to be the *selection probability* that governs the multivariate Bernoulli distribution used to generate the gate vector $\mathbf{m}$. Then, given $C_{\mathbf{R}}$, we can generate the gate vector from a multivariate Bernoulli distribution, which is denoted as $\mathbf{m} \sim \texttt{MultiBern}(\boldsymbol{\pi}; \mathbf{R})$, based on the correlated random variables $(U_1, \cdots, U_p)$ that preserve the correlation structure of the input features; formally, $m_k = 1$ if $U_k \leq \pi_k$ and $m_k = 0$ if $U_k > \pi_k$ for $k \in [p]$.

Using the correlation structure of the input features when generating the gate vector has the following two advantages: (i) during the Self-Supervision Phase, correlated gate vectors encourage the network not to rely on trivial signals from highly correlated features when solving the pretext tasks, which may lead to inefficient pre-training and sub-optimal downstream performance; (ii) during the Supervision Phase, correlated gate vectors encourage the network to select only the most rele-

vant features by increasing the chance that correlated features are selected jointly and thus compete against each other.

## 4.2 Self-Supervision Phase

Self-supervised learning uses unlabeled samples to automatically generate a (weak) supervisory signal by solving relevant pretext tasks in the absence of labeled data. The learned representations can then be used to solve downstream tasks. To this goal, we begin by defining two pretext tasks motivated by recent success in self-supervised learning for tabular data (Yoon et al., 2020). The pretext tasks to be jointly solved are: (i) reconstructing the original input $\mathbf{x}$ based on a randomly selected subset of input features $\tilde{\mathbf{x}}$, and (ii) simultaneously estimating the gate vector $\mathbf{m}$ that defined which features were selected and which features were not (i.e., replaced by their mean value).

To solve these pretext tasks, we formally define the encoder $f_\theta$ and two auxiliary network components $h_{\psi_1}, h_{\psi_2}$ that are temporarily employed during the Self-Supervision Phase:

- *Encoder*, $f_\theta : \mathcal{X}^p \rightarrow \mathcal{Z}$, that takes as input either the selected subset of features $\tilde{\mathbf{x}}$ or the original feature vector $\mathbf{x}$, and outputs latent representations $\mathbf{z} \in \mathcal{Z}$.
- *Feature vector estimator*, $h_{\psi_1} : \mathcal{Z} \rightarrow \mathcal{X}^p$, that takes $\mathbf{z} = f_\theta(\tilde{\mathbf{x}})$ as input and outputs a vector $\hat{\mathbf{x}}$ which is an estimate of the original input $\mathbf{x}$.
- *Gate vector estimator*, $h_{\psi_2} : \mathcal{Z} \rightarrow [0,1]^p$, that takes $\mathbf{z} = f_\theta(\tilde{\mathbf{x}})$ as input and outputs a vector $\hat{\mathbf{m}}$ which is a prediction of which features have been selected (i.e., $m_k = 1$) and which features have been replaced by the mean (i.e., $m_k = 0$).

When generating the subset of features $\tilde{\mathbf{x}}$, each feature is selected based on the multivariate Bernoulli distribution with an equal probability $\pi$, i.e., $\pi_k = \pi$ for $k \in [p]$, which is a hyper-parameter fixed throughout the Self-Supervision Phase. The adoption of an equal selection probability allows us to make no assumptions about the relative feature importance, which is not known *a priori*.

The purpose of the Self-Supervision Phase is to pre-train the encoder. As such, both estimators take as input the output of the same encoder $f_\theta$, which is the only component that will be retained in the Supervision Phase. Formally, given the latent representations $\mathbf{z} = f_\theta(\tilde{\mathbf{x}})$, the reconstructed input feature and the gate vector are defined as $\hat{\mathbf{x}} \triangleq h_{\psi_1} \circ f_\theta(\tilde{\mathbf{x}})$ and $\hat{\mathbf{m}} \triangleq h_{\psi_2} \circ f_\theta(\tilde{\mathbf{x}})$, respectively. The encoder and the two estimators are trained jointly based on the following objective function:

$$\underset{\theta, \psi_1, \psi_2}{\text{minimize}} \quad \mathbb{E}_{\mathbf{x} \sim p_X \, \mathbf{m} \sim p_M} \left[ \ell_X(\mathbf{x}, \hat{\mathbf{x}}) + \alpha \cdot \ell_M(\mathbf{m}, \hat{\mathbf{m}}) \right] \tag{5}$$

where $\alpha$ is a coefficient chosen to balance between the two pretext task losses: $\ell_X(\mathbf{x}, \hat{\mathbf{x}}) = \|\mathbf{x} - \hat{\mathbf{x}}\|_2^2$ and $\ell_M(\mathbf{m}, \hat{\mathbf{m}}) = -\sum_{k=1}^p m_k \log \hat{m}_k + (1 - m_k) \log(1 - \hat{m}_k)$.

Compared to previous work (Yoon et al., 2020), the correlated gating procedure in the proposed method prevents the pretext tasks from being solved by only exploiting trivial relationships among features by increasing the probability that highly-correlated features are masked simultaneously. Further, the pretext tasks directly mirror the Supervision Phase: in both phases, the input to the encoder is a subset of the features $\tilde{\mathbf{x}}$. This trains the encoder to produce better representations of partial feature sets which in turn prevents the model selection process in the Supervision Phase from overfitting to spurious, noisy features, benefiting feature selection.

## 4.3 Supervision Phase

During the Supervision Phase, the encoder, predictor, and the selection probability, which we refer to as the *feature selection network*, are jointly updated to solve the feature selection objective (3). The predictor is formally defined as follows:

- *Predictor*, $f_\phi : \mathcal{Z} \rightarrow \mathcal{Y}$, that takes as input the latent representations of the selected subset of features, i.e., $\mathbf{z} = f_\theta(\tilde{\mathbf{x}})$, and outputs predictions on the target outcome. Together with the encoder, the model selection function is given by $f = f_\phi \circ f_\theta$.

To account for the correlations between features, we generate gate vectors using a multivariate Bernoulli distribution. This is important since the correlated gate vectors encourage the network to select only the most relevant features by making multiple correlated features compete against

each other. However, solving (3) directly using such gate vectors is intractable since $p_M(\mathbf{m})$ has no differentiable closed-form expression. Instead, we adopt a continuous relaxation of the multivariate Bernoulli distribution (Wang & Yin, 2020) by applying the reparameterization trick to the correlated uniform random variables $(U_1, \cdots, U_p)$ that preserve the correlation structure $\mathbf{R}$.

Formally, given selection probability $\boldsymbol{\pi} = (\pi_1, \cdots, \pi_p)$ and the multivariate uniform random variables $(U_1, \cdots, U_p)$ from Gaussian copula $C_{\mathbf{R}}$, we can generate relaxed gate vector $\tilde{\mathbf{m}} = (\tilde{m}_1, \cdots, \tilde{m}_p) \in (0,1)^p$ based on the following reparameterization trick (Wang & Yin, 2020):

$$\tilde{m}_k = \sigma\left(\frac{1}{\tau}\big(\log \pi_k - \log(1-\pi_k) + \log U_k - \log(1-U_k)\big)\right) \tag{6}$$

where $\sigma(x) = (1 + \exp(-x))^{-1}$ is the sigmoid function. Such a relaxation is parameterized by $\boldsymbol{\pi}$, a pre-specified covariance matrix $\mathbf{R}$, and a temperature hyper-parameter $\tau \in (0, \infty)$. Similar to the Relaxed Bernoulli distribution (Maddison et al., 2017), (6) is differentiable with respect to $\boldsymbol{\pi}$.

In practice, the gate vector generation process in (6) can be handled as a deterministic transformation of samples from a standard multivariate Gaussian random variable. More specifically, we first draw samples from the multivariate Gaussian distribution $\mathbf{v} \sim \mathcal{N}(0, \mathbf{R})$ by $\mathbf{v} = (v_1, \cdots, v_p) = \mathbf{L}\boldsymbol{\epsilon}$ where $\boldsymbol{\epsilon} \sim \mathcal{N}(0, \mathbf{I})$ and $\mathbf{L}$ is a lower triangular matrix with positive diagonal entries, which is derived by the Cholesky factorization of the covariance matrix, i.e., $\mathbf{R} = \mathbf{L}\mathbf{L}^T$. Next, we generate correlated uniform random variables $u_k = \Phi(v_k)$ for $k \in [p]$. Finally, we can generate a relaxed multivariate Bernoulli variable $\tilde{m}_k \in (0,1)$ by plugging in $u_k$ and $\pi_k$ into (6).

Under the continuous relaxation, the regularization term in (3) that induces sparsity of selected features can be simply derived as $\mathbb{E}_{\mathbf{m} \sim p_M} \|\mathbf{m}\|_0 = \sum_{k=1}^p P(U_k \leq \pi_k) = \sum_{k=1}^p \pi_k$. Overall, we can rewrite our objective as

$$\underset{\theta, \phi, \boldsymbol{\pi}}{\text{minimize}} \quad \mathbb{E}_{\mathbf{x}, y \sim p_{XY}, \boldsymbol{\epsilon} \sim \mathcal{N}(0, \mathbf{I})}\left[\ell_Y\big(y, f_\phi \circ f_\theta(\tilde{\mathbf{m}} \odot \mathbf{x} + (1 - \tilde{\mathbf{m}}) \odot \mathbf{x})\big) + \beta \sum_{k=1}^p \pi_k\right]. \tag{7}$$

The model selection function $f = f_\phi \circ f_\theta$ and the selection probability $\boldsymbol{\pi}$ are updated jointly via gradient descent. Pseudo-code of SEFS can be found in Appendix A.

## 5 EXPERIMENTS

In this section, we evaluate the performance of SEFS and multiple feature selection methods using a synthetic and several real-world datasets. Further experiments together with detailed information regarding all experiments, benchmarks, and datasets can be found in the Appendix.

**Benchmarks.** We compare SEFS with 7 feature selection methods including 3 conventional methods and 4 state-of-the-art methods. The conventional methods include LASSO regularized linear/logistic model (**Lasso**, Tibshirani (1996)), extremely randomized tree ensembles (**Tree**, Geurts et al. (2006)), and Laplacian Score that preserves locality structure (**L-Score**, He et al. (2005)). State-of-the-art deep learning methods include Bayesian neural networks for feature selection (**BNNsel**, (Liang et al., 2018)), feature selection based on continuous relaxation (**STG**, Yamada et al. (2020)), unsupervised feature selection based on gated Laplacian (**DUFS**, Lindenbaum et al. (2020)). We also include an extension of STG to the semi-supervised setting by augmenting the loss with a reconstruction task that estimates the original features from the gated inputs (**STG (SS)**). Among the benchmarks, three are able to use both labeled and unlabeled samples: DUFS is fully unsupervised, while both L-Score and STG (SS) are semi-supervised approaches that explicitly use the label.

To highlight our novel two-step training process, we also include a variant of our method that is trained without employing the Self-Supervision Phase, which is denoted as **SEFS (no SS)**.

**Performance Metrics.** For the synthetic experiments, we use the true positive rate (TPR) to assess whether discovered features are truly relevant compared to the ground-truth. With real-world data, however, it is not possible to calculate the TPR since the ground truth relevance is often not known. Hence, we instead evaluate the prediction performance of the discovered features obtained by different feature selection methods. For classification tasks, we use the area under the receiver operating characteristic curve (AUROC) and area under the precision recall curve (AUPRC), and for regression tasks, mean squared error (MSE). This is an indirect way of assessing the relevance of the

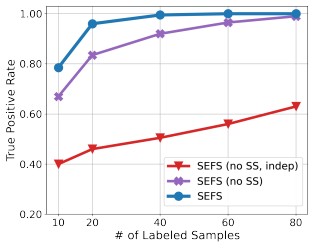 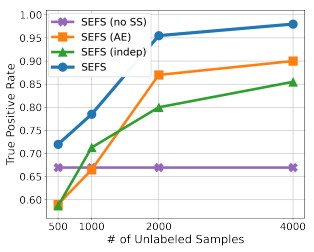

(a) TPR vs Labeled Samples     (b) TPR vs Unlabeled Samples

Figure 3: Results on the Block-Structured Noisy Two-Moons dataset. Average TPRs with with (a) varying numbers of labeled samples given $n_u=1000$ and (b) varying numbers of unlabeled samples given $n_l=10$.

Table 1: AUROC and AU-RPC (mean $\pm$ 95%-CI) given the learned representations for $n_l=20$, $n_u=1000$.

| Representation | AUROC |
|---|---|
| SEFS (AE) | 0.85±0.03 |
| SEFS (indep) | 0.88±0.03 |
| SEFS | **0.92±0.02** |

| Representation | AUPRC |
|---|---|
| SEFS (AE) | 0.81±0.04 |
| SEFS (indep) | 0.87±0.03 |
| SEFS | **0.91±0.03** |

discovered features on the target tasks: we first conduct feature selection based on each method and then train a multi-layer perceptron (MLP) to perform predictions on top of the *feature-selected* data. Training a separate MLP isolates the effect of having different model selections – such as linear models for Lasso, ensemble trees for Tree, and deep neural networks for BNNsel, STG and SEFS – and thereby provides a fair comparison of the discovered features.

**Experimental Setup.** For each dataset, we divide the available samples into labeled and unlabeled. Labeled samples are randomly split into training and testing sets. Within the training set, $n_l$ labeled samples from the training set are used to train the feature selection methods and the entire labeled samples are used to evaluate the discovered features. As a result, while features are selected based on limited sample sizes, a significantly larger number of labeled samples are used to validate the discovered features. Details of the data construction process is given in Appendix C.

**Implementation.** Implementation details, including hyper-parameters and the construction of co-variance matrix $\mathbf{R}$, together with details for all benchmarks can be found in Appendix B.

### 5.1 SYNTHETIC: BLOCK-STRUCTURED NOISY TWO-MOONS DATASET

To motivate and confirm our intuition of applying self-supervision and correlated gate vectors in feature selection, we begin by evaluating SEFS via a set of synthetic experiments that addresses two key challenges – i.e., limited labeled data and correlated features – in real-world scenarios.

**Dataset Description.** We consider the well-known 2-dimensional "Two-Moons" dataset with additive noise with variance $\sigma^2 = 0.1$. The outcome label (i.e., $\mathcal{Y} = \{0, 1\}$) is generated solely based on these 2 features (i.e., $x_1$ and $x_2$). We augment 8 auxiliary feature dimensions that are irrelevant to target outcomes by sampling from the standard Normal distribution. Finally, we introduce a correlated block-structure by augmenting each feature with 9 highly-correlated features (correlation: 0.94) by injecting small white noise (i.e., $\mathcal{N}(0, 0.3^2)$), yielding 100 features in total.

**Quantitative Analysis.** In Figure 3(a), we compare the average TPR with varying numbers of labeled samples $n_l = \{10, 20, 40, 60, 80\}$ while fixing the number of unlabeled samples $n_u = 1000$. We define $x_1$ (or $x_2$) as correctly discovered if and only if $x_1$ (or $x_2$) has the first or second highest feature importance. SEFS benefits from learning the data structure using unlabeled samples and provides significant gain over all baselines (see Appendix D.1 for a full comparison). The majority of benchmarks struggle to identify $x_1$ in particular due to $x_2$ being more discriminative than $x_1$, and therefore noisy features that are correlated with $x_2$ are often selected (Table S.4). We highlight the benefit of the correlated gating procedure by comparing SEFS (no SS) and SEFS (no SS, indep), a variant that uses the independent Bernoulli distribution to generate gate vectors. Our novel gating procedure significantly improves the TPR even in the fully supervised setting (Figure 3(a)).

We further investigate the utility of the Self-Supervision Phase in SEFS. To this goal, we introduce two counterparts that replace our self-supervised training with (i) a conventional auto-encoder (SEFS (AE)) and (ii) a variant of our model with an uncorrelated gating procedure using independent Bernoulli random variables (SEFS (indep)). In all cases, we employ the same feature selection process as SEFS to isolate the impact of self-supervision. In Figure 3(b), we evaluate these variants in terms of the TPR with a varying number of unlabeled samples $n_u = 200, 500, 1000, 2000$, and 4000 while fixing the number of labeled samples $n_l = 10$. (i.e., 5 for each label). Our method significantly outperforms the other variants, improving the TPR over SEFS (no SS) even with a relative

small number of unlabeled samples. Moreover, we further evaluate the quality of the learned representation by assessing the discriminative performance of the learned representations given only the ground-truth features (i.e., setting $m_1 = m_2 = 1$ and $m_k = 0$ for $k \notin \{1, 2\}$). As shown in Table 1, the proposed gating procedure in SEFS helps the encoder to learn meaningful structure and results in the most performant representations, further validating the self-supervised training procedure.

## 5.2 CLINICAL: UKCF DATASET

**Dataset Description.** The UK Cystic Fibrosis registry (UKCF)[3] records annual follow-ups for 6,754 adults over the period 2008–2015. We include patients with observations in at least three consecutive years. Each patient is associated with $p$=245 clinical variables (11 static and $3 \times 78$ time-varying features), including demographics, genetic mutations, clinical tests, and therapeutic management. We set respiratory failure (death or lung transplant) in the next 5 years as the label.

Table 2: AUROC and AUPRC (mean $\pm$ 95%-CI) for $|\mathcal{S}| = 10$ discovered features for the UKCF dataset.

| Methods | AUROC | AUPRC |
|---|---|---|
| Lasso | 0.767±0.054 | 0.409±0.083 |
| Tree | 0.807±0.036 | 0.463±0.069 |
| BNNsel | 0.650±0.051 | 0.269±0.069 |
| STG | 0.781±0.048 | 0.440±0.082 |
| DUFS | 0.799±0.039 | 0.398±0.062 |
| L-Score | 0.668±0.010 | 0.213±0.017 |
| STG (SS) | 0.810±0.036 | 0.477±0.043 |
| SEFS (no SS) | 0.785±0.044 | 0.412±0.083 |
| **SEFS** | **0.846±0.013** | **0.532±0.027** |

**Quantitative Results.** In Table 2, we compare the prediction performance of feature selection methods with $n_l = 32$, $n_u = 4754$ and $|\mathcal{S}| = 10$. As expected, SEFS benefits from learning the data structure utilizing the unlabeled samples and discovers features that provide significantly greater discriminative performance over all other methods (Table 2). Fully supervised feature selection methods struggle to discover relevant features with only a few labeled samples. The benefit of self-supervision can be clearly seen by comparing SEFS with SEFS (no SS). Our results also highlight that fully unsupervised feature selection methods (i.e., L-Score and DUFS) struggle to discover relevant features as these methods cannot select features that effectively distinguish the target outcomes without the guidance of label information.

**Qualitative Results.** Two interesting features identified by SEFS but that were not discovered by other methods were PI Allele 1 and 2 that are related to pancreatic functions. More specifically, pancreatic function studies have demonstrated that (types of) mutations in the CFTR gene, which causes CF, are highly associated with dysfunction in organ systems and the pathology of CF patients (Sosnay et al., 2013; Gibson-Corley et al., 2016). Further details of the most frequently discovered features and supporting evidence of their clinical relevance can be found in Appendix D.2.

## 5.3 PROTEOMICS: CCLE DATASET

**Dataset Description.** We study the response of heterogeneous cancer cell lines to 11 different drugs where the goal is to identify proteins associated with the cell line response based on proteomic measurements from the Cancer Cell Line Encyclopedia (CCLE, Barretina et al. (2012)). CCLE is a small dataset containing 899 cancer cell lines (i.e., samples) described by $p$=196 protein expressions. The real-valued drug response (i.e., $\mathcal{Y} = \mathbb{R}$) is available for 458 samples and is missing for the remaining 441 samples (thus unlabeled). To benefit from self-supervised learning, we integrate the RPPA measurements on 7,329 samples from The Cancer Genome Atlas (TCGA)[4], creating overall 7,770 unlabeled samples (i.e, $n_u = 7770$). While the two datasets are not generated from the same distribution, we expect some benefit from learning basic correlations among features, even if not all correlations from CCLE are preserved.

Table 3: Frequently discovered features by SEFS for Panobinostat (CCLE). Features in blue are **not** chosen by SEFS (no SS). Supporting references provided in Appendix D.3.

| Rank | Proteins | Ref. |
|---|---|---|
| 1 | Caveolin-1 | ✓ |
| 2 | YAP-pS127 | ✓ |
| 3 | PRAS40-pT246 | ✓ |
| 4 | VHL | ✓ |
| 5 | Src-pY416 | ✓ |
| 6 | TAZ | ✓ |
| 7 | 14-3-3-$\beta$ | ✓ |
| 8 | Fibronectin | – |
| 9 | GSK3-pS9 | ✓ |
| 10 | MSH2 | ✓ |

**Quantitative Results.** In Figure 4, we compare the ranking of SEFS and the benchmarks across 11 drugs ($n_l = 20$, $n_u = 7770$, $|\mathcal{S}| = 10$). SEFS is the best performing method for 9 drugs and is always in the top 3 (median rank: 1, average rank: 1.36). Despite the majority of unlabeled data originating from a different source, SEFS outperforms SEFS (no SS) in every experiment. While

---

[3] https://www.cysticfibrosis.org.uk/
[4] https://www.cancer.gov/

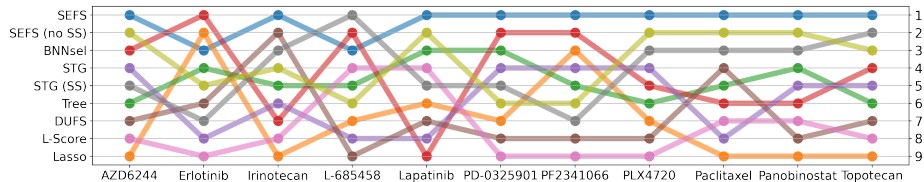

Figure 4: Ranking of the average prediction performance for $|\mathcal{S}| = 10$ discovered features across different drugs in the CCLE dataset. (Rank: 1 = best, 9 = worst.)

Table 4: AUROC and AURPC (mean $\pm$ 95%-CI) given $|\mathcal{S}|$=20 discovered features for the PBMC dataset.

| Methods | AUROC | AUPRC |
|---|---|---|
| Lasso | 0.703±0.032 | 0.705±0.041 |
| Tree | 0.767±0.044 | 0.788±0.047 |
| BNNsel | 0.676±0.035 | 0.686±0.035 |
| STG | 0.761±0.066 | 0.777±0.071 |
| DUFS | 0.734±0.029 | 0.742±0.034 |
| L-Score | 0.699±0.009 | 0.688±0.009 |
| STG (SS) | 0.794±0.065 | 0.814±0.069 |
| SEFS (no SS) | 0.768±0.055 | 0.783±0.057 |
| **SEFS** | **0.884±0.013** | **0.901±0.013** |

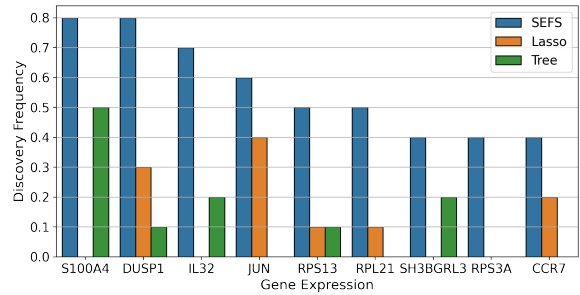

Figure 5: Comparison of discovered features using Lasso, Tree, and SEFS for the PBMC dataset with $n_l$=20.

we would expect further gain from more similar unlabeled data, our results highlight that there is potential benefit even when unlabeled data is only partially related to the labeled samples.

**Qualitative Results.** Now we have demonstrated the superior predictive performance of the features discovered using SEFS, we seek to validate these features in the scientific literature. Here, we focus our analysis on *panobinostat*, a histone deacetylase (HDAC) inhibitor used in the treatment of several cancers. We found strong supporting evidence for 9 of the top 10 ranked features selected by SEFS (Table 3, see Appendix D.3 for full details). In addition, the remaining feature (fibronectin) is strongly implicated in cancer, but we did not find literature specifically relating it to panobinostat or HDAC-mediated pathways. We provide supporting evidence for other drugs in Appendix D.3.

## 5.4 TRANSCRIPTOMICS: PBMC DATASET

**Dataset Description.** We focus on distinguishing sub-populations of T-cells (i.e., $\mathcal{Y} = \{0,1\}$), namely naive and regulatory T-cells, from purified populations of peripheral blood monocytes (PBMCs)[5] based on transcriptomic measurements (i.e., mRNA sequence). The PBMC dataset consists of 20,742 samples described by $p$=19256 protein-coding genes. We primarily used $n_l = 20$, $n_u = 15557$, with the remaining 5,165 labeled samples used to evaluate the discovered features.

**Quantitative and Qualitative Results.** Similarly to the performance on other datasets, SEFS significantly outperforms all benchmarks (Table 4). Figure S.4 displays only features selected with a frequency $\geq 0.4$. It is immediately apparent that only SEFS consistently identifies the same features, while Lasso and Tree typically select different features, demonstrating the robustness of our approach. We provide supporting evidence for the importance of these features in Appendix D.4.

## 6 CONCLUSION

In this paper, we proposed SEFS, a self-supervised feature selection framework that is able to leverage the abundant quantities of unlabeled data that are often available, achieving state-of-the-art performance when limited labeled data is available. Using synthetic data, we motivate and confirm our intuition as to why self-supervision and our novel gating procedure can benefit feature selection methods. Through experiments on multiple real-world datasets, we validate that our model discovers features that provide superior prediction performance and corroborate the vast majority of the discovered features with supporting medical and scientific literature. As with all feature selection methods, the discovered features based on SEFS should be further experimentally verified or evaluated by a domain expert prior to deployment in practice.

---

[5] https://support.10xgenomics.com/single-cell-gene-expression/datasets

## ACKNOWLEDGMENTS

We thank anonymous reviewers as well as members of the vanderschaar-lab for many insightful comments and suggestions. CL was supported through the IITP grant funded by the Korea government (MSIT) (No. 2021-0-01341, AI Graduate School Program, CAU). FI and MvdS are supported by the National Science Foundation (NSF), grant number 1722516. MvdS is additionally supported by the Office of Naval Research (ONR).

## ETHICS STATEMENT

Like all methods for selecting features or providing the importance of features from observational data, SEFS relies on an assumption (i.e., the important features subset should be enough to provide good prediction power). Consequently, all identified features should undergo additional evaluation or verification by domain experts prior to deployment in practice. In this work, we highlight direct/indirect impact of our method through experiments on real-world clinical and omics datasets including the UKCF, CCLE, and PBMC datasets. The use of these datasets was in accordance with the guidance of the respective data providers and domain experts.

## REPRODUCIBILITY STATEMENT

The source code of SEFS and the synthetic dataset are provided in the Supplementary Material and is also available at `https://github.com/chl8856/SEFS`. All details of our method and the experimental setup are included in either the main manuscript or the Appendix: a description of the method can be found in Section 4, pseudo-code for SEFS is described in Appendix A, Appendix B provides implementation details for both SEFS and the benchmarks, Appendix C provides details about the real-world datasets.

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

# A PSEUDO-CODE OF SEFS

SEFS is trained via a two-step training procedure. We provide pseudo-codes for the Self-Supervision Phase in Algorithm 1 and for the Supervision Phase in Algorithm 2.

The source code for SEFS is available in `https://github.com/chl8856/SEFS`.

---

**Algorithm 1** Pseudo-code for the Self-Supervision Phase of SEFS

---

**Input:** Dataset $\mathcal{D}_u = \{\mathbf{x}^i\}_{i=1}^{n_u}$, coefficient $\alpha$, selection probability $\pi$,
  mini-batch size $n_{mb}$, learning rate $\eta$
**Output:** SEFS parameter $\theta$

Initialize parameters $(\theta, \psi_1, \psi_2)$
Compute the mean value: $\bar{\mathbf{x}} = \frac{1}{n_u} \sum_{i=1}^{n_u} \mathbf{x}^i$

Compute the correlation matrix: $\mathbf{R}$ where $R_{kj} = \frac{|C_{kj}|}{\sqrt{C_{kk} \cdot C_{jj}}}$ and $C_{kj} = \frac{\sum_{i=1}^{n_u} (x_k^i - \bar{x}_k^i)(x_j^i - \bar{x}_j^i)}{n_u - 1}$

**repeat**
  Sample a mini-batch of $n_{mb}$ unlabeled data samples: $\{\mathbf{x}^i\}_{i=1}^{n_{mb}} \sim \mathcal{D}_u$
  **for** $i = 1, \cdots, n_{mb}$ **do**
    Sample gate vector: $\mathbf{m}^i \sim \texttt{MultiBern}(\pi; \mathbf{R})$
    Generate feature subset: $\tilde{\mathbf{x}}^i \leftarrow \mathbf{m} \odot \mathbf{x}^i + (1 - \mathbf{m}) \odot \bar{\mathbf{x}}$
    Estimate feature vector: $\hat{\mathbf{x}}^i \leftarrow h_{\psi_1} \circ f_\theta(\tilde{\mathbf{x}}^i)$
    Estimate gate vector: $\hat{\mathbf{m}}^i \leftarrow h_{\psi_2} \circ f_\theta(\tilde{\mathbf{x}}^i)$
  **end for**
  Update the encoder parameter $\theta$:

$$\theta \leftarrow \theta - \eta \nabla_\theta \left( \frac{1}{n_{mb}} \sum_{i=1}^{n_{mb}} \ell_X(\mathbf{x}^i, \hat{\mathbf{x}}^i) + \alpha \cdot \ell_M(\mathbf{m}^i, \hat{\mathbf{m}}^i) \right)$$

  Update the feature vector estimator parameter $\psi_1$:

$$\psi_1 \leftarrow \psi_1 - \eta \nabla_{\psi_1} \left( \frac{1}{n_{mb}} \sum_{i=1}^{n_{mb}} \ell_X(\mathbf{x}^i, \hat{\mathbf{x}}^i) \right)$$

  Update the gate vector estimator parameter $\psi_2$:

$$\psi_2 \leftarrow \psi_2 - \eta \nabla_{\psi_2} \left( \frac{1}{n_{mb}} \sum_{i=1}^{n_{mb}} \alpha \cdot \ell_M(\mathbf{m}^i, \hat{\mathbf{m}}^i) \right)$$

**until** convergence

---

# B IMPLEMENTATION DETAILS

The hyper-parameters of SEFS and those of the benchmarks are chosen via a grid search. For all methods, we use 20% of the overall training set as the validation set, which will then be unseen for training feature selection methods with chosen hyper-parameters.

Once feature selection methods are trained utilizing a small subset of the overall training set as illustrated in Figure S.1, we train MLPs (with 3 layers and 100 nodes) with feature-selected data. As described in Section 5, training a separate MLP isolates the effect of having different model selections and thereby provides a fair comparison among the discovered features.

It is worth highlighting that generating correlated gate vectors can be computationally burdensome for high-dimensional data due to the matrix multiplication in Algorithm 3 and 4. To avoid such an issue, we first apply thresholding (e.g., 0.7 for the PBMC dataset) to the correlation matrix $\mathbf{R}$, group

---

**Algorithm 2** Pseudo-code for the Supervision Phase of SEFS

---

**Input:** Dataset $\mathcal{D}_l = \{\mathbf{x}^i, y^i\}_{i=1}^{n_l}$, pre-trained encoder $\theta$, coefficient $\beta$,
   mini-batch size $n_{mb}$, learning rate $\eta$
**Output:** SEFS parameters $(\theta, \phi, \boldsymbol{\pi})$

Initialize parameters $(\phi, \boldsymbol{\pi})$
Compute the mean value: $\bar{\mathbf{x}} = \frac{1}{n_u} \sum_{i=1}^{n_u} \mathbf{x}^i$

Compute the correlation matrix: $\mathbf{R}$ where $R_{kj} = \frac{|C_{kj}|}{\sqrt{C_{kk} \cdot C_{jj}}}$ and $C_{kj} = \frac{\sum_{i=1}^{n_u}(x_k^i - \bar{x}_k^i)(x_j^i - \bar{x}_j^i)}{n_u - 1}$

**repeat**
   Sample a mini-batch of $n_{mb}$ labeled data samples: $\{\mathbf{x}^i, y^i\}_{i=1}^{n_{mb}} \sim \mathcal{D}_l$
   **for** $i = 1, \cdots, n_{mb}$ **do**
      Sample gate vector: $\tilde{\mathbf{m}}^i \sim$ `Relaxed-MultiBern`$(\boldsymbol{\pi}; \mathbf{R})$
      Generate feature subset: $\tilde{\mathbf{x}}^i \leftarrow \tilde{\mathbf{m}} \odot \mathbf{x}^i + (1 - \tilde{\mathbf{m}}) \odot \bar{\mathbf{x}}$
      Predict outcome given the feature subset: $\hat{y}^i \leftarrow f_\phi \circ f_\theta(\tilde{\mathbf{x}}^i)$
   **end for**
   Fine-tune the encoder parameter $\theta$:

$$\theta \leftarrow \theta - \eta \nabla_\theta \left( \frac{1}{n_{mb}} \sum_{i=1}^{n_{mb}} \ell_Y(y^i, \hat{y}^i) \right)$$

   Update the predictor parameter $\phi$:

$$\phi \leftarrow \phi - \eta \nabla_\phi \left( \frac{1}{n_{mb}} \sum_{i=1}^{n_{mb}} \ell_Y(y^i, \hat{y}^i) \right)$$

   Update the selection probability parameter $\boldsymbol{\pi}$:

$$\boldsymbol{\pi} \leftarrow \boldsymbol{\pi} - \eta \nabla_{\boldsymbol{\pi}} \left( \frac{1}{n_{mb}} \sum_{i=1}^{n_{mb}} \ell_Y(y^i, \hat{y}^i) + \beta \sum_{k=1}^{p} \pi_k \right)$$

**until** convergence

---

**Algorithm 3** Pseudo-code for `MultiBern`$(\boldsymbol{\pi}; \mathbf{R})$

---

**Input:** selection probability $\boldsymbol{\pi}$, correlation matrix $\mathbf{R}$
**Output:** correlated gate vector $\mathbf{m}$
Draw a standard normal sample: $\epsilon \sim \mathcal{N}(0, \mathbf{I})$
Compute $\mathbf{L} =$ `Cholesky-Decomposition`$(\mathbf{R})$
Generate a multivariate Gaussian vector $\mathbf{v} = \mathbf{L}\epsilon$
**for** $k = 1, \cdots, p$ **do**
   Apply element-wise Gaussian CDF: $u_k = \Phi(v_k)$
   Generate correlated gate: $m_k = \mathbb{1}(u_k \leq \pi_k)$
**end for**

---

features based on the agglomerative clustering[6] using the correlation matrix as the similarity measure, and then conduct block-wise matrix multiplication. Overall, the generated gates for features within the same group will maintain the correlation structure. This significantly reduces the computational complexity in Algorithm 3 and 4 as it scales quadratically with respect to the largest block size (i.e., the number of features grouped in the largest block). Thus, if the largest block size remains the same, the complexity of generating the correlated gate vectors will only increase linearly with the feature dimension (since the number of blocks would increase linearly).

---

[6]Implemented using Python package `scikit-learn`

---

**Algorithm 4** Pseudo-code for `Relaxed-MultiBern`$(\boldsymbol{\pi}; \mathbf{R})$ (Wang & Yin, 2020)

---

**Input:** selection probability $\boldsymbol{\pi}$, correlation matrix $\mathbf{R}$, temperature $\tau$
**Output:** correlated gate vector $\tilde{\mathbf{m}}$
Draw a standard normal sample: $\epsilon \sim \mathcal{N}(0, \mathbf{I})$
Compute $\mathbf{L} = $ `Cholesky-Decomposition`$(\mathbf{R})$
Generate a multivariate Gaussian vector $\mathbf{v} = \mathbf{L}\epsilon$
**for** $k = 1, \cdots, p$ **do**
    Apply element-wise Gaussian CDF: $u_k = \Phi(v_k)$
    Apply reparameterization trick (Maddison et al., 2017):

$$\tilde{m}_k = \sigma\left(\frac{1}{\tau}\big(\log \pi_k - \log(1 - \pi_k) + \log u_k - \log(1 - u_k)\big)\right)$$

    where $\sigma(x) = \frac{1}{1+\exp(-x)}$.
**end for**

---

Throughout the experiments, training SEFS takes approximately 30 minutes to 1 hour for each phase (similar to that of STG) on a single GPU machine[7].

## B.1 SEFS

The overall training process of SEFS consists of 4 different network components that are implemented as neural networks: (i) encoder $f_\theta$, (ii) predictor $f_\phi$, (iii), feature vector estimator $h_{\phi_1}$, and (iv) gate vector estimator $h_{\phi_2}$. We use a fully-connected network as the baseline architecture for all the network components.

For the Semi-Supervision Phase, coefficients $(\alpha, \pi)$ and the hyper-parameters – including the number of hidden units, the number of nodes, and the number of layers – of the encoder, the feature vector estimator, and the gate vector estimator are chosen utilizing a grid search. We choose hyper-parameter values that achieve the lowest validation loss.

For the Supervision Phase, coefficient $\beta$ and the hyper-parameters – including the number of nodes and the number of layers – of the predictor are chosen utilizing a grid search (note that the hyper-parameters of the encoder is fixed after the Semi-Supervision Phase). We choose hyper-parameter values that achieve the best validation performance by using the outputs of the predictor (i.e., $\hat{y} = f_\phi \circ f_\theta(\mathbf{x})$).

The permitted values of coefficients $(\alpha, \beta, \pi)$ and the hyper-parameters are listed in Table S.1.

## B.2 BENCHMARKS

Throughout the experiments, we compared SEFS with feature selection methods ranging from conventional approaches – such as Lasso (Tibshirani, 1996) and Tree (Geurts et al., 2006) – to the state-of-the-art approaches – such as STG (Yamada et al., 2020) and DUFS (Lindenbaum et al., 2020). Further descriptions of the benchmarks and implementation details are as follows:

- Lasso[8]: Lasso is a well-known embedded method whose objective is to minimize the prediction loss (i.e., cross-entropy loss for classification tasks and mean squared error for regression tasks) while enforcing the $l_1$-penalty to achieve sparsity among input features (Tibshirani, 1996). We implement Lasso regression for regression tasks and Logistic regression with $l_1$-penalty for classification tasks. The hyper-parameter to control the input sparsity is chosen from $\{0.001, 0.01, 0.1, 1, 10, 100\}$.

- Tree[9]: ExtraTree (Geurts et al., 2006) is an ensemble of decision trees that has become a popular choice for classification and regression tasks for tabular data. We use the feature importance

---

[7]The specification of the machine is: CPU – Intel Core i7-8700K, GPU – NVIDIA GeForce GTX 1080Ti, and RAM – 64GB DDR4
[8]Implemented using Python package `scikit-learn`
[9]Implemented using Python package `scikit-learn`

Table S.1: Hyper-parameters of SEFS.

| Block | Set of Hyper-Parameters |
|---|---|
| Initialization | Xavier (Glorot & Bengio, 2010) |
| Optimization | Adam (Kingma & Ba, 2015) |
| Mini-batch size* | 32 |
| Non-linearity | ReLu |
| **Semi-Supervision Phase** | |
| Learning rate ($\eta$) | $\{0.0001, 0.001, 0.01\}$ |
| No. of hidden units | $\{10, 30, 50, 100\}$ |
| No. of layers | $\{1, 2, 3\}$ |
| No. of nodes | $\{10, 30, 50, 100, 300, 500\}$ |
| Coeff. $\alpha$ | $\{0.01, 0.1, 1.0, 10, 100\}$ |
| Coeff. $\pi$ | $\{0.2, 0.4, 0.6, 0.8\}$ |
| **Supervision Phase** | |
| Temperature $\tau$ | 1.0 |
| Dropout | 0.3 |
| Learning rate ($\eta_1, \eta_2$) | $\{0.0001, 0.001, 0.01\}$ |
| No. of layers | $\{1, 2, 3\}$ |
| No. of nodes | $\{10, 30, 50, 100, 300, 500\}$ |
| Coeff. $\beta$ | $\{0.01, 0.1, 1.0, 10, 100\}$ |

*We used $n_l$ as the mini-batch size for the cases with
the number of labeled samples smaller than 32, i.e., $\min(n_l, 32)$.

obtained by ExtraTree to conduct feature selection. The number of trees and the maximum depth are chosen from $\{10, 50, 100, 300, 500\}$ and $\{1, 2, 3, 4\}$, respectively.

- L-Score[10]: The Laplacian Score (L-Score) (He et al., 2005) is an unsupervised (filter) feature selection method that quantifies the importance of input features by the ability to preserve local structure of the data, which is captured by the Laplacian matrix. We further modify the affinity score between a pair of labeled samples following the description in Sheikhpour et al. (2017), i.e., the affinity score becomes 1 if the two samples have the same label and 0 otherwise. We construct the Laplacian matrix based on the Gaussian kernel using both labeled and unlabeled samples with 10-nearest neighbors and setting the kernel bandwidth as the median Euclidean distance.

- BNNsel[11]: BNNsel is an embedded feature selection method based on Bayesian neural networks that overcomes the non-differentiability in the selection process by utilizing MCMC sampling (Liang et al., 2018). The number of hidden units is chosen among $\{3$ (default value), $5, 10, 30, 50\}$ with a fixed prior probability 0.025 (default value).

- STG[12]: STG is an embedded feature selection method using deep neural networks that overcomes the non-differentiable process of selecting feature subsets via continuous relaxation using Gaussian random variables (Yamada et al., 2020). We use a fully-connected network as the baseline architecture where the number of nodes and the number of layers are chosen among $\{10, 30, 50, 100, 300, 500\}$ and $\{1,2,3,4,5\}$, respectively. The hyper-parameter to control the sparsity of input features is chosen among $\{0.001, 0.01, 0.1, 1, 10, 100\}$.

- STG (SS): We propose an extension of STG to the semi-supervised setting. To this goal, we introduce a reconstruction task in addition to the prediction task of the original STG. More specifically, we utilize an auxiliary network, i.e., decoder, which reconstructs the original input features given the gated features by minimizing the reconstruction loss. The overall network is trained based on the combination of the prediction loss and the reconstruction loss. (Here, we adopt a hyper-parameter $\alpha \in \{1., 0.1, 0.01, 0.001\}$ to balance the two losses). We use a fully-connected network as the baseline architecture where the number of nodes and the number of layers are chosen among $\{10, 30, 50, 100, 300, 500\}$ and $\{1,2,3,4,5\}$, respectively. The hyper-

---

[10]We explicitly implement L-Score for semi-supervised setting based on the description in Sheikhpour et al. (2017)

[11]https://rdrr.io/cran/BNN/man/BNNsel.html

[12]https://github.com/runopti/stg

Table S.2: Summary statistics of the dataset.

| Data Source | UKCF | PBMC | CCLE | | | | | |
|---|---|---|---|---|---|---|---|---|
| | | | *Irino* | *L* | *PLX* | *Pano* | *AZD6244 Erlo Pacli* | *Lapa PD PF Topo* |
| **Data Type** | clinical | transcriptomics | proteomics | | | | | |
| **Target Outcome** | resp. failure | cell-type class | drug response | | | | | |
| **Feature Dimension** | 285 | 19256 | 196 | | | | | |
| **No. Labeled Samples (FS–Train)** | 32 | 20 | 15 | 23 | 25 | 27 | 29 | 29 |
| **No. Labeled Samples (Eval.–Train)** | 1000 | 1296 | 146 | 223 | 225 | 227 | 229 | 229 |
| **No. Labeled Samples (Eval.–Test)** | 1000 | 3889 | 146 | 223 | 225 | 227 | 228 | 229 |
| **No. Unlabeled Samples** | 4754 | 15557 | 7770 | 7770 | 7770 | 7770 | 7770 | 7770 |

*Irino: Irinotecan, PLX: PLX4720, L: L-685458, Pano: Panobinostat, AZD: AZD6244, Erlo: Erlotinib, Pacli: Paclitaxel*
*Lapa: Lapatinib, PD: PD-0325901, PF: PF2341066, Topo :Topotecan*

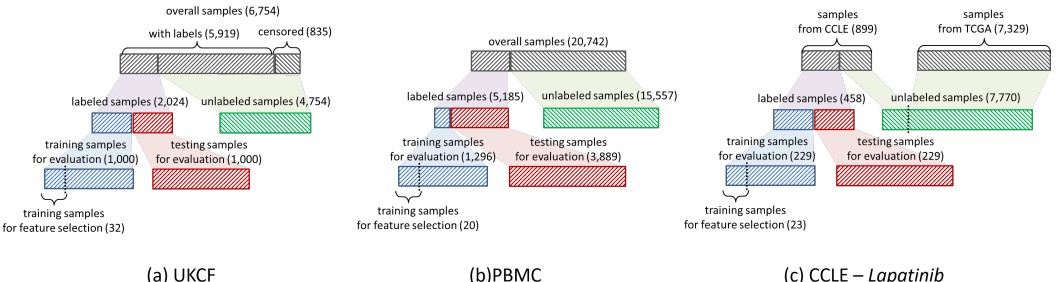

(a) UKCF  (b)PBMC  (c) CCLE – *Lapatinib*

Figure S.1: Illustration of dataset splits for (a) UKCF, (b) PBMC, and (c) CCLE – *Lapatinib* datasets.

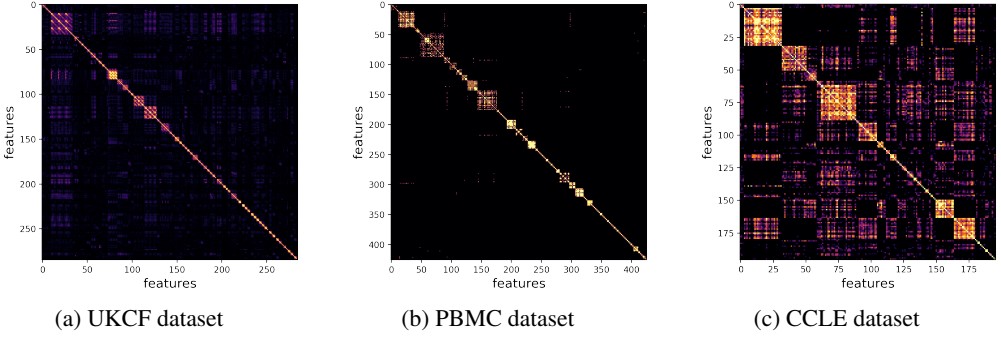

(a) UKCF dataset  (b) PBMC dataset  (c) CCLE dataset

Figure S.2: The correlation structure of (a) UKCF, (b) PBMC, and (c) CCLE datasets. For the PBMC dataset, the number of features are reduced after thresholding.

parameter to control the sparsity of input features is chosen among $\{0.001, 0.01, 0.1, 1, 10, 100\}$.

- DUFS[13]: DUFS is an unsupervised feature selection method that improves the Laplacian score (He et al., 2005) by replacing it with a gated variant computed on a subset of features (Lindenbaum et al., 2020). We use the parameter-free version of DUFS which construct the Laplacian matrix with 2-nearest neighbors and setting the kernel bandwidth as the median Euclidean distance. Both labeled and unlabeled samples are used for training.

## C  DETAILED DATA DESCRIPTIONS

In Section 5 of the manuscript, we evaluate SEFS and the benchmarks with multiple healthcare datasets from three different sources, i.e., UKCF (clinical), CCLE (proteomics), and PBMC (transcriptomics). The summary statistics of these datasets are provided in Table S.2.

---

[13]https://github.com/Ofirlin/DUFS

Throughout the experiments on the real-world healthcare and omics datasets, we consider a practical scenario for feature selection under low labeled data regime where only a small number of labeled samples are available while having a sufficient amount of unlabeled sample. To this goal, we construct datasets based on the following steps: First, we obtain enough unlabeled samples by utilizing samples without labels that are readily present in the original data (e.g., UKCF dataset has censored patients and CCLE dataset has samples without drug response), treating a portion of labeled samples as unlabeled (e.g., UKCF and PBMC datasets), and by integrating samples from different sources (e.g., CCLE dataset). Second, we randomly split the labeled samples into training and testing sets. Here, both the training and the testing sets are used for evaluating discovered features from different feature selection methods. This assures that the discovered features generalize well to unseen samples. Third, we randomly select a small subset of training samples for training different feature selection methods. The overall data construction steps are illustrated in Figure S.1.

**Correlation Structure.** The correlation structure of the datasets are illustrated in Figure S.2. We construct the correlation matrix $\mathbf{R}$ as defined in Algorithm 1 and 2: formally, each element of $\mathbf{R}$ is given as $R_{kj} = \frac{|C_{kj}|}{\sqrt{C_{kk} \cdot C_{jj}}}$ where $C_{kj} = \frac{\sum_{i=1}^{n_u}(x_k^i - \bar{x}_k^i)(x_j^i - \bar{x}_j^i)}{n_u - 1}$. It is worth highlighting that the PBMC dataset contains many features that have low correlation with other features; thus, after thresholding at 0.7, only 426 features with high correlation remain.

To quantify the amount of collinearity, we calculated the variance inflation factor (VIF), which is an index that measures how much the variance (the square of the estimate's standard deviation) of an estimated regression coefficient is increased because of collinearity. While there is not a strict cut-off indicating collinearity, a general rule of thumb is that VIFs exceeding 10 are signs of serious multicollinearity (Menard, 2001; Vittinghoff et al., 2011; James et al., 2013). We note this is a conservative cut-off, with guidelines suggesting that VIFs greater than 5 (Menard, 2001; Vittinghoff et al., 2011) indicate considerable collinearity. In Table S.3, we provide the VIFs of features in each dataset; here the VIFs for the PBMC dataset are not available due to the computational complexity. As seen in the table, there are strong signs of multicollinearity in both datasets with the VIFs for 23.5% of the features in the UKCF dataset and all the features in the CCLE dataset exceeding 10.

Table S.3: VIFs for the tested real-world datasets.

| Datasets | Mean | Min | Max | VIF > 10 |
|----------|-------|-------|--------|----------|
| UKCF | 11.33 | 1.04 | 391.84 | 23.50% |
| CCLE | 80.81 | 15.00 | 462.97 | 100% |

# D  ADDITIONAL EXPERIMENTS

## D.1  SYNTHETIC: BLOCK-STRUCTURED NOISY TWO-MOONS DATASET

**Quantitative Results.** Table S.4 shows the average TPRs for all benchmarks. As can be seen in the table, SEFS outperforms all benchmarks, with the majority of methods struggling to identify Feature 1 in particular. This is due to Feature 2 being more discriminative than Feature 1 and therefore the noisy features that are correlated with Feature 2 are often selected. It is worth highlighting that L-Score, DUFS, and STG (SS) fail to discover discriminative features in almost all cases due to the correlation structure of the data that cannot be addressed by the similarity metrics or reconstruction employed by these methods.

## D.2  CLINICAL: UKCF DATASET

**Quantitative Results.** Figure S.3 displays features discovered with a frequency $\geq 0.4$, where we consider a feature is discovered (for all three feature selection methods) if that feature is within the top 20 highest feature importance. It is immediately apparent that SEFS consistently identifies the same features, while Lasso and Tree typically select different features, demonstrating the robustness of our approach.

Table S.4: The average TPRs on the Two-Moons dataset.

| Methods | $(n_\ell=20, n_u=1000)$ | | | $(n_\ell=80, n_u=1000)$ | | |
|---|---|---|---|---|---|---|
| | Feature 1 | Feature 2 | Average | Feature 1 | Feature 2 | Average |
| Lasso | 0.45 | 0.98 | 0.72 | 0.77 | 1.00 | 0.89 |
| Tree | 0.02 | 0.46 | 0.24 | 0.05 | 0.88 | 0.47 |
| STG | 0.14 | 0.96 | 0.55 | 0.21 | 1.00 | 0.61 |
| DUFS | 0.02 | 0.03 | 0.03 | 0.01 | 0.02 | 0.02 |
| L-Score | 0.00 | 0.00 | 0.00 | 0.00 | 0.00 | 0.00 |
| STG (SS) | 0.09 | 1.00 | 0.55 | 0.09 | 0.94 | 0.52 |
| SEFS (no SS) | 0.68 | 0.99 | 0.84 | 0.98 | 1.00 | 0.99 |
| **SEFS** | **0.92** | **1.00** | **0.96** | **1.00** | **1.00** | **1.00** |

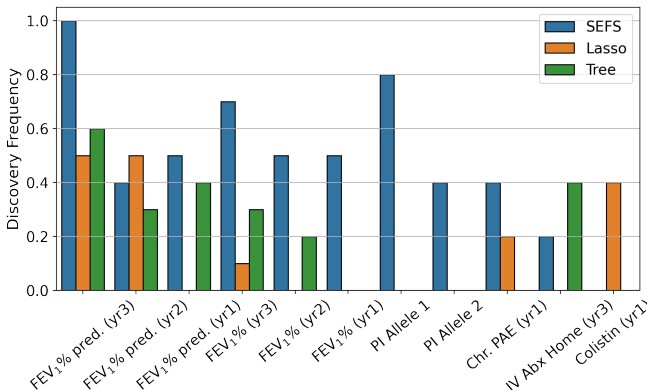

Figure S.3: Comparison of discovered features based on Lasso, Tree, and SEFS for the UKCF dataset with $n_l = 32$.

**Qualitative Results.** Other than PI Allele 1 and 2 that are related to pancreatic functions as described in Section 5.2, SEFS consistently discovered features related to the lung function scores (i.e., $FEV_1\%$ pred and $FEV_1$). It is worth highlighting that, features related to the lung function scores are frequently selected despite their relatively high correlation. We conjecture that the trajectory of the lung functions scores measured at different times (with the highest frequency of the latest measurement) plays an important role for predicting the respiratory failure as supported in Taylor-Robinson et al. (2012); Adler & Liou (2016).

### D.3  PROTEOMICS: CCLE DATASET

**Quantitative Results.** In Table S.5, we compare the prediction performance of 5 and 10 discovered features (i.e., $|\mathcal{S}| = 5$ and 10) for the 11 drugs – AZD6244, Erlotinib, Irinotecan, L-685458, Lapatinib, PD-0325901, PF2341066, PLX4720, Paclitaxel, Panobinostat, and Topotecan – reported in the manuscript. Despite the majority of unlabeled data originating from a different source, our method benefits from learning the underlying data structure from unlabeled samples. SEFS consistently displays improvements (except for Erlotinib with $|\mathcal{S}| = 5$ and 10, L-685458 with $|\mathcal{S}| = 5$, and PF2341066 with $|\mathcal{S}| = 5$ and 10) in performance for different drugs and varying numbers of discovered features, outperforming SEFS (no SS) in every experiment. While we would expect further gain from more similar unlabeled data, our results highlight the effectiveness of self-supervision even when the unlabeled data is only partially related to the labeled samples.

**Qualitative Results.** In the main manuscript, we validated the features identified by SEFS for panobinostat in the scientific literature. In Table S.6, we provide details of the references for panobinostat and also provide supporting evidence for the selected features for lapatinib and irinotecan.

Lapatinib is a drug used primarily to treat breast cancer, as well as other solid tumours. It is a dual tyrosine kinase inhibitor which inhibits the epidermal growth factor receptor (EGFR) and human epidermal growth factor receptor 2 (HER2) receptors. In total, we found supporting literature for 8

Table S.5: Comparison of the MSE (mean ± 95%-CI) given discovered features for the CCLE dataset. (Lower the better.)

| Methods | AZD6244 | | Erlotinib | | Irinotecan | | L-685458 | |
|---|---|---|---|---|---|---|---|---|
| | $\|\mathcal{S}\| = 5$ | $\|\mathcal{S}\| = 10$ | $\|\mathcal{S}\| = 5$ | $\|\mathcal{S}\| = 10$ | $\|\mathcal{S}\| = 5$ | $\|\mathcal{S}\| = 10$ | $\|\mathcal{S}\| = 5$ | $\|\mathcal{S}\| = 10$ |
| Lasso | 1.444±0.03 | 1.494±0.03 | 0.426±0.02 | 0.426±0.02 | 1.432±0.03 | 1.416±0.03 | 0.305±0.01 | 0.315±0.01 |
| Tree | 1.389±0.04 | 1.426±0.04 | 0.407±0.02 | 0.434±0.02 | 1.385±0.03 | 1.340±0.03 | 0.293±0.01 | 0.309±0.01 |
| BNNsel | 1.394±0.04 | 1.412±0.04 | **0.400±0.01** | **0.413±0.01** | 1.364±0.03 | 1.354±0.03 | 0.288±0.01 | 0.305±0.01 |
| STG | 1.385±0.04 | 1.415±0.04 | 0.419±0.01 | 0.446±0.02 | 1.399±0.03 | 1.347±0.03 | 0.295±0.01 | 0.316±0.01 |
| DUFS | 1.427±0.04 | 1.443±0.04 | 0.420±0.01 | 0.442±0.01 | 1.373±0.03 | 1.323±0.03 | 0.311±0.01 | 0.343±0.01 |
| L-Score | 1.396±0.03 | 1.480±0.04 | 0.433±0.02 | 0.447±0.01 | 1.439±0.04 | 1.384±0.04 | 0.298±0.01 | 0.309±0.01 |
| STG (SS) | 1.393±0.04 | 1.416±0.04 | 0.411±0.01 | 0.444±0.01 | 1.409±0.04 | 1.327±0.04 | **0.281±0.01** | **0.295±0.01** |
| SEFS (no SS) | 1.376±0.04 | 1.409±0.04 | 0.408±0.01 | 0.440±0.01 | 1.414±0.03 | 1.335±0.04 | 0.284±0.01 | 0.312±0.01 |
| **SEFS** | **1.363±0.03** | **1.403±0.03** | 0.404±0.02 | 0.428±0.02 | **1.358±0.04** | **1.291±0.04** | 0.282±0.01 | 0.307±0.01 |

| Methods | Lapatinib | | PD-0325901 | | PF2341066 | | PLX4720 | |
|---|---|---|---|---|---|---|---|---|
| | $\|\mathcal{S}\| = 5$ | $\|\mathcal{S}\| = 10$ | $\|\mathcal{S}\| = 5$ | $\|\mathcal{S}\| = 10$ | $\|\mathcal{S}\| = 5$ | $\|\mathcal{S}\| = 10$ | $\|\mathcal{S}\| = 5$ | $\|\mathcal{S}\| = 10$ |
| Lasso | 0.450±0.01 | 0.459±0.02 | 2.216±0.05 | 2.210±0.05 | 0.365±0.01 | 0.365±0.01 | 0.407±0.02 | 0.420±0.02 |
| Tree | 0.412±0.02 | 0.440±0.02 | 2.196±0.04 | 2.178±0.05 | 0.345±0.01 | 0.370±0.01 | 0.397±0.02 | 0.416±0.02 |
| BNNsel | 0.457±0.02 | 0.474±0.02 | 2.176±0.05 | 2.178±0.05 | 0.347±0.01 | 0.363±0.01 | 0.397±0.02 | 0.414±0.02 |
| STG | 0.438±0.02 | 0.464±0.02 | 2.210±0.06 | 2.179±0.06 | 0.350±0.01 | 0.369±0.01 | 0.403±0.01 | 0.412±0.01 |
| DUFS | 0.436±0.01 | 0.460±0.01 | 2.186±0.05 | 2.233±0.04 | 0.359±0.01 | 0.376±0.01 | 0.414±0.01 | 0.423±0.01 |
| L-Score | 0.439±0.01 | 0.451±0.01 | 2.435±0.04 | 2.269±0.05 | 0.349±0.01 | 0.384±0.01 | 0.422±0.01 | 0.429±0.01 |
| STG (SS) | 0.435±0.01 | 0.456±0.01 | 2.184±0.05 | 2.183±0.05 | 0.344±0.01 | 0.373±0.01 | 0.409±0.02 | 0.410±0.02 |
| SEFS (no SS) | 0.394±0.02 | 0.424±0.02 | 2.171±0.06 | 2.194±0.06 | 0.345±0.01 | 0.371±0.01 | 0.396±0.01 | 0.400±0.02 |
| **SEFS** | **0.392±0.02** | **0.405±0.02** | **2.168±0.05** | **2.163±0.06** | **0.344±0.01** | **0.360±0.01** | **0.396±0.01** | **0.396±0.02** |

| Methods | Paclitaxel | | Panobinostat | | Topotecan | |
|---|---|---|---|---|---|---|
| | $\|\mathcal{S}\| = 5$ | $\|\mathcal{S}\| = 10$ | $\|\mathcal{S}\| = 5$ | $\|\mathcal{S}\| = 10$ | $\|\mathcal{S}\| = 5$ | $\|\mathcal{S}\| = 10$ |
| Lasso | 1.689±0.03 | 1.689±0.03 | 0.651±0.01 | 0.631±0.02 | 1.533±0.02 | 1.530±0.02 |
| Tree | 1.633±0.03 | 1.626±0.03 | 0.547±0.02 | 0.531±0.02 | 1.441±0.04 | 1.431±0.03 |
| BNNsel | 1.636±0.03 | 1.629±0.03 | 0.574±0.02 | 0.552±0.02 | 1.436±0.03 | 1.415±0.03 |
| STG | 1.654±0.03 | 1.650±0.04 | 0.573±0.02 | 0.539±0.01 | 1.433±0.03 | 1.425±0.03 |
| DUFS | 1.667±0.04 | 1.625±0.04 | 0.607±0.02 | 0.575±0.02 | 1.443±0.03 | 1.439±0.03 |
| L-Score | 1.675±0.03 | 1.639±0.03 | 0.624±0.01 | 0.554±0.01 | 1.501±0.02 | 1.491±0.02 |
| STG (SS) | 1.618±0.03 | 1.622±0.03 | 0.541±0.02 | 0.522±0.02 | 1.408±0.03 | 1.393±0.03 |
| SEFS (no SS) | 1.614±0.04 | 1.604±0.03 | 0.522±0.02 | 0.506±0.01 | 1.416±0.03 | 1.407±0.03 |
| **SEFS** | **1.605±0.04** | **1.587±0.04** | **0.512±0.02** | **0.496±0.01** | **1.405±0.03** | **1.389±0.03** |

of the top 10 ranked features selected by SEFS (Table S.6b), with five such features not proposed by SEFS (no SS). The other two features selected by SEFS are both established cancer biomarkers with evidence of their importance in breast cancer, but we did not find literature specifically relating them to Lapatinib.

Irinotecan is a chemotherapy agent used primarily to treat colon cancer in addition to small cell lung cancer. All of the top 10 features identified by SEFS had supporting scientific literature for their impact on the mechanism and effectiveness of irintecan or colon cancer (Table S.6c), of which six were not proposed by SEFS (no SS).

## D.4   TRANSCRIPTOMICS: PBMC DATASET

**Qualitative Results.** In Figure S.4, we compare the frequency of the discovered features based on SEFS with that of the discovered features based on Lasso and Tree. The figure displays only the features that were selected with a frequency equal to or greater than 0.4 by one of the methods. It is worth highlighting that SEFS (no SS) does not discover any feature with frequency equal to or greater than 0.4, which shows the importance of the Self-Supervision Phase. The lack of consistency of discovering features suggests that Lasso and Tree could be overfit to spurious relations in different splits of the data. Only 2 features are discovered at least 40% of the time by either Lasso or Tree (one feature each), compared to 9 for SEFS.

SEFS almost always discovers *S100A4* and *IL32* as relevant features. Both of these genes have been shown to be only expressed, or preferentially expressed, in certain types of T-cells – *S100A4*: Weatherly et al. (2015) and *IL32*: Goda et al. (2006) – which strongly validates the selection of these features. These features are also the most frequent features discovered by Tree, but with much lower frequency, while Lasso does not discover either feature, despite their high relevance. Similarly, while *DUSP1* (also known as *MKP-1*) is selected by Lasso, Tree, and SEFS, only our

Table S.6: Frequently discovered proteins (features) by SEFS for the CCLE dataset. Features in blue were **not** proposed by SEFS (no SS).

(a) Panobinostat

| Rank | Proteins | Ref. |
|------|----------|------|
| 1 | Caveolin-1 | (Deb et al., 2016) |
| 2 | YAP-pS127 | (Heinemann et al., 2015) |
| 3 | PRAS40-pT246 | (Gallagher et al., 2018) |
| 4 | VHL | (Kalac et al., 2011) |
| 5 | Src-pY416 | (Kostyniuk et al., 2002) |
| 6 | TAZ | (Lee et al., 2017) |
| 7 | 14-3-3-$\beta$ | (Wang et al., 2000) |
| 8 | Fibronectin | – |
| 9 | GSK3-pS9 | (Rahmani et al., 2014) |
| 10 | MSH2 | (Li et al., 2020) |

(b) Lapatinib

| Rank | Proteins | Ref. |
|------|----------|------|
| 1 | Caveolin-1 | (Qian et al., 2019) |
| 2 | HER2-pY1248 | (Medina & Goodin, 2008) |
| 5 | VHL | – |
| 6 | mTOR | (Gayle et al., 2012) |
| 7 | 53BP1 | (Li et al., 2012) |
| 8 | N-Ras | (Galiè, 2019) |
| 9 | 14-3-3-$\beta$ | – |
| 8 | Claudin-7 | (Constantinou et al., 2018) |
| 9 | Rab25 | (Cheng et al., 2013) |
| 10 | GSK3_pS9 | (Duda et al., 2020) |

(c) Irinotecan

| Rank | Proteins | Ref. |
|------|----------|------|
| 1 | Lck | (Harashima et al., 2001) |
| 2 | Stathmin | (Peng et al., 2010) |
| 3 | $\beta$-Catenin | (Saifo et al., 2010) |
| 4 | Bcl-2 | (Gupta et al., 2007) |
| 5 | Src_pY416 | (Petitprez & Larsen, 2013) |
| 6 | Smad4 | (Wong et al., 2020) |
| 7 | YAP_pS127 | (Noguchi et al., 2018) |
| 8 | Bcl-xL | (Lee et al., 2019) |
| 9 | PI3K-p85 | (Koizumi et al., 2005) |
| 10 | E-Cadherin | (Bendardaf et al., 2019) |

method discovers this feature in the majority of the experiments, despite its importance for T-cell activation and function (Zhang et al., 2009).

In addition, we also find evidence of the importance of *JUN* (Riera-Sans & Behrens, 2007), *SH3BGRL3* (Deng et al., 2006), *CCR7* (Schneider et al., 2007), and ribosomal genes *RPS13*, *RPL21*, and *RPS3A* (Procaccini et al., 2016).

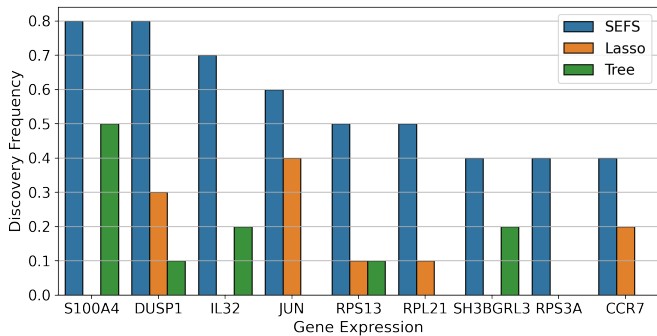

Figure S.4: Comparison of discovered features using Lasso, Tree, and SEFS for the PBMC dataset with $n_l$=20 (i.e., 10 labeled samples for each label).

## D.5 CORRELATED VS INDEPENDENT GATE VECTOR GENERATION IN FEATURE SELECTION

In this subsection, we investigate how the correlated gate vectors using a multivariate Bernoulli distribution in SEFS improves the feature selection process over independently generated gate vectors. To this goal, in Figure S.5, we provide trajectories of selection probabilities for the ground-truth feature $x_2$ and its block-correlated noisy features with respect to the number of training iterations. Note that the results are generated only utilizing the Supervision Phase. As claimed in the manuscript, the correlated gate vectors encourage the network to select only the most relevant feature by making

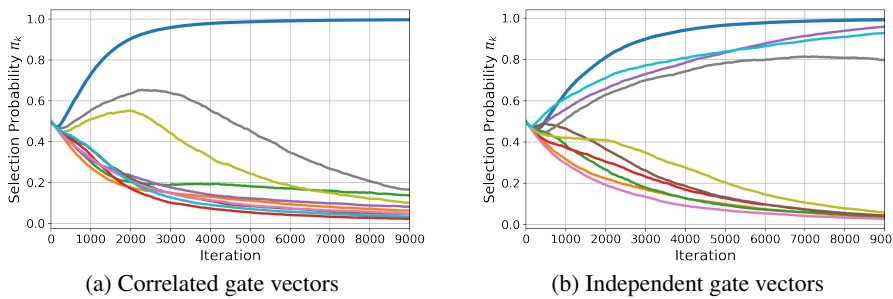

(a) Correlated gate vectors           (b) Independent gate vectors

Figure S.5: The trajectory of $\pi_k$'s with respect to the number of iterations for the ground-truth relevant feature $x_2$ (in the blue line) and its block-correlated noisy features (in other colors) using (a) correlated gate vectors using a multivariate Bernoulli distribution and (b) independent gate vectors using a independent Bernoulli distribution.

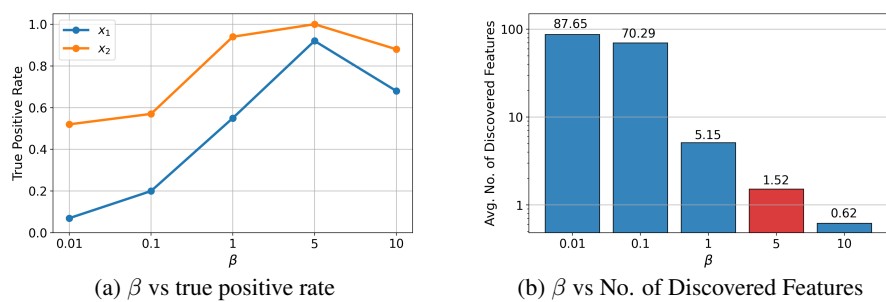

(a) $\beta$ vs true positive rate           (b) $\beta$ vs No. of Discovered Features

Figure S.6: Comparison of (a) the TPR performance and (b) the number of discovered features with $\pi_k > 0.5$, with respect to different values of $\beta$, respectively.

multiple correlated features compete against each other. This is highlighted in Figure S.5. However, when independent gate vectors are used, it is less likely to make correlated features to compete each other and increase the selection probabilities of the correlated noisy features because these features are still informative about the target compared to the augmented features – i.e., $(x_3, \cdots, x_{10})$ – and their block-correlated features that are purely noisy.

### D.6 SENSITIVITY ANALYSIS ON COEFFICIENT $\beta$

In this subsection, we provide sensitivity analysis using the Block-Structured Noisy Two-Moons dataset to see the effect of coefficient $\beta$ that controls the sparsity of the selected features. Figure S.6 shows (a) the true discovery rate as defined in Section 5.1 – where we define $x_1$ (or $x_2$) as correctly discovered if and only if $x_1$ (or $x_2$) has the first or the second highest feature importance – and (b) the number of features discovered whose selection probability is higher than 0.5 (i.e., $\pi_k > 0.5$). Here, multiple instances of SEFSwith different $\beta$'s are trained utilizing $n_l = 20$ labeled samples (i.e., 10 labeled samples for each label) and pre-trained with $n_u = 1000$. The results are averaged over 100 iterations as described in Section 5.1 of the manuscript.

Figure S.6(a) shows that SEFS trained with setting $\beta = 5, 10$ provides the highest TPR. However, as can be seen in Figure S.6(b), setting the coefficient too small (here, $\beta \leq 1$) will end up discovering too many irrelevant features as if they were relevant – i.e., having many features whose selection probability is above a certain level – and will mislead the discovery of relevant features. Contrarily, setting the coefficient too high (here, $\beta = 10$) can restrict the selection process and may fail to identify relevant features.

APPENDIX REFERENCES

Frederick R. Adler and Theodore G. Liou. The dynamics of disease progression in cystic fibrosis. *PLOS ONE*, 11(6):1–17, 06 2016.

Riyad Bendardaf, Fatemeh Saheb Sharif-Askari, Narjes Saheb Sharif-Askari, Kari Syrjänen, and Seppo Pyrhönen. Cytoplasmic e-cadherin expression is associated with higher tumour level of vegfa, lower response rate to irinotecan-based treatment and poorer prognosis in patients with metastatic colorectal cancer. *Anticancer Research*, 39(4):1953–1957, 2019. ISSN 0250-7005. doi: 10.21873/anticanres.13305. URL https://ar.iiarjournals.org/content/39/4/1953.

Kw Cheng, R. Agarwal, S. Mitra, and Gb Mills. Rab25 small gtpase mediates secretion of tumor necrosis factor receptor superfamily member 11b (osteoprotegerin) protecting cancer cells from effects of trail. *Journal of genetic syndromes & gene therapy*, 4:1000153, 2013. doi: 10.4172/2157-7412.1000153.

Chloe Constantinou, Savvas Papadopoulos, Eirini Karyda, Athanasios Alexopoulos, Niki Agnanti, Anna Batistatou, and Haris Harisis. Expression and clinical significance of claudin-7, pdl-1, pten, c-kit, c-met, c-myc, alk, ck5/6, ck17, p53, egfr, ki67, p63 in triple-negative breast cancer-a single centre prospective observational study. *In vivo (Athens, Greece)*, 32(2):303–311, 2018.

Moonmoon Deb, Dipta Sengupta, Swayamsiddha Kar, Sandip Kumar Rath, Subhendu Roy, Goutam Das, and Samir Kumar Patra. Epigenetic drift towards histone modifications regulates cav1 gene expression in colon cancer. *Gene*, 581(1):75–84, 2016. doi: https://doi.org/10.1016/j.gene.2016.01.029.

Y. J. Deng, Z. X. Huang, C. J. Zhou, J. W. Wang, Y. You, Z. Q. Song, M. M. Xiang, B. Y. Zhong, and F. Hao. Gene profiling involved in immature cd4+ t lymphocyte responsible for systemic lupus erythematosus. *Molecular Immunology*, 43(9):1497–1507, 2006.

Przemysław Duda, Shaw M. Akula, Stephen L. Abrams, Linda S. Steelman, Alberto M. Martelli, Lucio Cocco, Stefano Ratti, Saverio Candido, Massimo Libra, Giuseppe Montalto, Melchiorre Cervello, Agnieszka Gizak, Dariusz Rakus, and James A. McCubrey. Targeting gsk3 and associated signaling pathways involved in cancer. *Cells*, 9(5):1110, Apr 2020. ISSN 2073-4409. doi: 10.3390/cells9051110.

Mirco Galiè. Ras as supporting actor in breast cancer. *Frontiers in oncology*, 9:1199–1199, Nov 2019. ISSN 2234-943X. doi: 10.3389/fonc.2019.01199.

Stuart J Gallagher, Dilini Gunatilake, Kimberley A Beaumont, Danae M Sharp, Jessamy C Tiffen, Anja Heinemann, Wolfgang Weninger, Nikolas K Haass, James S Wilmott, Jason Madore, Peter M Ferguson, Helen Rizos, and Peter Hersey. Hdac inhibitors restore braf-inhibitor sensitivity by altering pi3k and survival signalling in a subset of melanoma. *International Journal of Cancer*, 142(9):1926–1937, 2018. doi: https://doi.org/10.1016/j.cellsig.2019.03.014.

Sylvia S. Gayle, Samuel L. M. Arnold, Ruth M. O'Regan, and Rita Nahta. Pharmacologic inhibition of mtor improves lapatinib sensitivity in her2-overexpressing breast cancer cells with primary trastuzumab resistance. *Anti-cancer agents in medicinal chemistry*, 12(2):151–162, Feb 2012. ISSN 1875-5992. doi: 10.2174/187152012799015002.

Pierre Geurts, Damien Ernst, and Louis Wehenkel. Extremely randomized trees. *Machine Learning*, 63(1):3–42, 2006.

X. Glorot and Y. Bengio. Understanding the difficulty of training deep feedforward neural networks. *In Proceedings of the 13th International Conference on Artificial Intelligence and Statistics (AISTATS 2010)*, 2010.

Chiho Goda, Taisuke Kanaji, Sachiko Kanaji, Go Tanaka, Kazuhiko Arima, Shigeaki Ohno, and Kenji Izuhara. Involvement of IL-32 in activation-induced cell death in T cells. *International Immunology*, 18(2):233–240, 2006. doi: 10.1093/intimm/dxh339.

Vinay Gupta, Yuzhuang S Su, Christian G Samuelson, Leonard F Liebes, Marc C Chamberlain, Florence M Hofman, Axel H Schönthal, and Thomas C Chen. Irinotecan: a potential new chemotherapeutic agent for atypical or malignant meningiomas. *Journal of neurosurgery*, 106(3):455—462, March 2007. ISSN 0022-3085. doi: 10.3171/jns.2007.106.3.455. URL https://doi.org/10.3171/jns.2007.106.3.455.

Nanae Harashima, Koji Tanaka, Teruo Sasatomi, Kanako Shimizu, Yoshiaki Miyagi, Akira Yamada, Mayumi Tamura, Hideaki Yamana, Kyogo Itoh, and Shigeki Shichijo. Recognition of the lck tyrosine kinase as a tumor antigen by cytotoxic t lymphocytes of cancer patients with distant metastases. *European Journal of Immunology*, 31(2):323–332, 2001. doi: https://doi.org/10.1002/1521-4141(200102)31:2⟨323::AID-IMMU323⟩3.0.CO;2-0.

X. He, D. Cai, and P. Niyogi. Laplacian score for feature selection. *In Proceedings of the 18th Conference on Neural Information Processing Systems (NIPS 2005)*, 2005.

Anja Heinemann, Carleen Cullinane, Ricardo De Paoli-Iseppi, James S. Wilmott, Dilini Gunatilake, Jason Madore, Dario Strbenac, Jean Y. Yang, Kavitha Gowrishankar, Jessamy C. Tiffen, Rab K. Prinjha, Nicholas Smithers, Grant A. McArthur, Peter Hersey, and Stuart J. Gallagher. Combining bet and hdac inhibitors synergistically induces apoptosis of melanoma and suppresses akt and yap signaling. *Oncotarget*, 6(25):21507–21521, Aug 2015. ISSN 1949-2553. URL https://pubmed.ncbi.nlm.nih.gov/26087189.

G. James, D. Witten, T. Hastie, and R. Tibshirani R. *An Introduction to Statistical Learning: With Applications in R*. Springer, 2013.

Matko Kalac, Luigi Scotto, Enrica Marchi, Jennifer Amengual, Venkatraman E. Seshan, Govind Bhagat, Netha Ulahannan, Violetta V. Leshchenko, Alexis M. Temkin, Samir Parekh, Benjamin Tycko, and Owen A. O'Connor. HDAC inhibitors and decitabine are highly synergistic and associated with unique gene-expression and epigenetic profiles in models of DLBCL. *Blood*, 118(20):5506–5516, 11 2011. ISSN 0006-4971. doi: 10.1182/blood-2011-02-336891. URL https://doi.org/10.1182/blood-2011-02-336891.

D. P. Kingma and J. Ba. Adam: A method for stochastic optimization. *In Proceedings of the 3rd International Conference on Learning Representations (ICLR 2015)*, 2015.

Naoki Koizumi, Etsuro Hatano, Takashi Nitta, Masaharu Tada, Nobuko Harada, Kojiro Taura, Iwao Ikai, and Yasuyuki Shimahara. Blocking of pi3k/akt pathway enhances apoptosis induced by sn-38, an active form of cpt-11, in human hepatoma cells. *Int J Oncol*, 26(5):1301–1306, May 2005. doi: 10.3892/ijo.26.5.1301. URL https://doi.org/10.3892/ijo.26.5.1301.

Calley L. Kostyniuk, Scott M. Dehm, Danielle Batten, and Keith Bonham. The ubiquitous and tissue specific promoters of the human src gene are repressed by inhibitors of histone deacetylases. *Oncogene*, 21(41):6340–6347, Sep 2002. ISSN 1476-5594. doi: 10.1038/sj.onc.1205787. URL https://doi.org/10.1038/sj.onc.1205787.

Boah Lee, Jeong A. Min, Abdullateef Nashed, Sang-Ok Lee, Jae Cheal Yoo, Seung-Wook Chi, and Gwan-Su Yi. A novel mechanism of irinotecan targeting mdm2 and bcl-xl. *Biochemical and Biophysical Research Communications*, 514(2):518–523, 2019. ISSN 0006-291X. doi: https://doi.org/10.1016/j.bbrc.2019.04.009. URL https://www.sciencedirect.com/science/article/pii/S0006291X1930631X.

Wen-Ying Lee, Pin-Cyuan Chen, Wen-Shin Wu, Han-Chung Wu, Chun-Hsin Lan, Yen-Hua Huang, Chia-Hsiung Cheng, Ku-Chung Chen, and Cheng-Wei Lin. Panobinostat sensitizes kras-mutant non-small-cell lung cancer to gefitinib by targeting taz. *International Journal of Cancer*, 141(9):1921–1931, 2017. doi: https://doi.org/10.1002/ijc.30888. URL https://onlinelibrary.wiley.com/doi/abs/10.1002/ijc.30888.

Shiqin Li, Bingbing Shi, Xinli Liu, and Han-Xiang An. Acetylation and deacetylation of DNA repair proteins in cancers. *Frontiers in Oncology*, 10, Oct 2020. doi: 10.3389/fonc.2020.573502. URL https://doi.org/10.3389/fonc.2020.573502.

Xiaoyan Li, Bing Xu, Meena S. Moran, Yuhan Zhao, Peng Su, Bruce G. Haffty, Changshun Shao, and Qifeng Yang. 53bp1 functions as a tumor suppressor in breast cancer via the inhibition of nf-$\kappa$b through mir-146a. *Carcinogenesis*, 33(12):2593–2600, Dec 2012. ISSN 0143-3334. doi: 10.1093/carcin/bgs298. URL https://doi.org/10.1093/carcin/bgs298.

Faming Liang, Qizhai Li, and Lei Zhou. Bayesian neural networks for selection of drug sensitive genes. *Journal of the American Statistical Association*, 113(523):955–972, 2018.

O. Lindenbaum, U. Shaham, J. Svirsky, E. Peterfreund, and Y. Kluger. Differentiable unsupervised feature selection based on a gated laplacian. *arXiv preprint arXiv:2007.04728*, 2020.

C. J. Maddison, A. Mnih, and Y. W. Teh. The concrete distribution: A continuous relaxation of discrete random variables. *In Proceedings of the 5th International Conference on Learning Representations (ICLR 2017)*, 2017.

Patrick J. Medina and Susan Goodin. Lapatinib: A dual inhibitor of human epidermal growth factor receptor tyrosine kinases. *Clinical Therapeutics*, 30(8):1426–1447, 2008.

S. Menard. *Applied Logistic Regression Analysis*. SAGE Publications, Inc, 2001.

Satoshi Noguchi, Akira Saito, and Takahide Nagase. Yap/taz signaling as a molecular link between fibrosis and cancer. *International journal of molecular sciences*, 19(11):3674, Nov 2018. ISSN 1422-0067. doi: 10.3390/ijms19113674.

Xing-Chen Peng, Feng-Ming Gong, Meng Wei, Xi Chen, Ye Chen, Ke Cheng, Feng Gao, Feng Xu, Feng Bi, and Ji-Yan Liu. Proteomic analysis of cell lines to identify the irinotecan resistance proteins. *Journal of Biosciences*, 35(4):557–564, 2010.

Amelie Petitprez and Annette K. Larsen. Irinotecan resistance is accompanied by upregulation of egfr and src signaling in human cancer models. *Current Pharmaceutical Design*, 19(5):958–964, 2013. URL http://www.eurekaselect.com/node/105574/article.

Claudio Procaccini, Fortunata Carbone, Dario Di Silvestre, Francesca Brambilla, Veronica De Rosa, Mario Galgani, Deriggio Faicchia, Gianni Marone, Donatella Tramontano, Marco Corona, Carlo Alviggi, Antonio Porcellini, Antonio La Cava, Pierluigi Mauri, and Giuseppe Matarese. The proteomic landscape of human ex vivo regulatory and conventional t cells reveals specific metabolic requirements. *Immunity*, 44(2):406–421, Feb 2016. ISSN 1097-4180. doi: 10.1016/j.immuni.2016.01.028.

Xian-Ling Qian, Yi-Hang Pan, Qi-Yuan Huang, Yu-Bo Shi, Qing-Yun Huang, Zhen-Zhen Hu, and Li-Xia Xiong. Caveolin-1: a multifaceted driver of breast cancer progression and its application in clinical treatment. *OncoTargets and therapy*, 12:1539–1552, Feb 2019. ISSN 1178-6930. doi: 10.2147/OTT.S191317.

Mohamed Rahmani, Mandy Mayo Aust, Elisa C. Benson, LaShanale Wallace, Jonathan Friedberg, and Steven Grant. Pi3k/mtor inhibition markedly potentiates hdac inhibitor activity in nhl cells through bim- and mcl-1–dependent mechanisms in vitro and in vivo. *Clinical Cancer Research*, 20(18):4849–4860, 2014. doi: 10.1158/1078-0432.CCR-14-0034. URL https://clincancerres.aacrjournals.org/content/20/18/4849.

Lluís Riera-Sans and Axel Behrens. Regulation of $\alpha\beta/\gamma\delta$ t cell development by the activator protein 1 transcription factor c-jun. *The Journal of Immunology*, 178(9):5690–5700, 2007.

Maher S. Saifo, Donald R. Rempinski Jr., Youcef M. Rustum, and Rami G. Azrak. Targeting the oncogenic protein beta-catenin to enhance chemotherapy outcome against solid human cancers. *Molecular cancer*, 9:310–310, Dec 2010. doi: 10.1186/1476-4598-9-310.

Martin A. Schneider, Josef G. Meingassner, Martin Lipp, Henrietta D. Moore, and Antal Rot. Ccr7 is required for the in vivo function of cd4+ cd25+ regulatory t cells. *Journal of Experimental Medicine*, 204(4):735–745, Mar 2007. ISSN 0022-1007. doi: 10.1084/jem.20061405. URL https://doi.org/10.1084/jem.20061405.

Razieh Sheikhpour, Mehdi Agha Sarram, Sajjad Gharaghani, and Mohammad Ali Zare Chahooki. A survey on semi-supervised feature selection methods. *Pattern Recognition*, 64:141–158, 2017.

David Taylor-Robinson, Margaret Whitehead, Finn Diderichsen, Hanne Vebert Olesen, Tania Pressler, Rosalind L Smyth, and Peter Diggle. Understanding the natural progression in %fev1 decline in patients with cystic fibrosis: a longitudinal study. *Thorax*, 67(10):860–866, 2012.

Robert Tibshirani. Regression shrinkage and selection via the lasso. *Journal of the Royal Statistical Society: Series B (Methodological)*, 58(1):267–288, 1996.

E. Vittinghoff, D. V. Glidden, S. C. Shiboski, and C. E. McCulloch. *Regression Methods in Biostatistics: Linear, Logistic, Survival, and Repeated Measures Models*. 2011.

A. H. Wang, M. J. Kruhlak, J. Wu, N. R. Bertos, M. Vezmar, B. I. Posner, D. P. Bazett-Jones, and X. J. Yang. Regulation of histone deacetylase 4 by binding of 14-3-3 proteins. *Molecular and cellular biology*, 20(18):6904–6912, Sep 2000.

Xi Wang and Junming Yin. Relaxed multivariate bernoulli distribution and its applications to deep generative models. *In Proceedings of the 26th Conference on Uncertainty in Artificial Intelligence (UAI 20)*, pp. 425–432, 2020.

Kathleen Weatherly, Marie Bettonville, David Torres, Arnaud Kohler, Stanislas Goriely, and Michel Y. Braun. Functional profile of s100a4-deficient t cells. *Immunity, Inflammation and Disease*, 3(4):431–444, 2015. doi: https://doi.org/10.1002/iid3.85.

Chen Khuan Wong, Arthur W. Lambert, Sait Ozturk, Panagiotis Papageorgis, Delia Lopez, Ning Shen, Zaina Sen, Hamid M. Abdolmaleky, Balázs Győrffy, Hui Feng, and Sam Thiagalingam. Targeting rictor sensitizes smad4-negative colon cancer to irinotecan. *Molecular Cancer Research*, 18(3):414–423, 2020. ISSN 1541-7786. doi: 10.1158/1541-7786.MCR-19-0525. URL `https://mcr.aacrjournals.org/content/18/3/414`.

Y. Yamada, O. Lindenbaum, S. Negahban, and Y. Kluger. Feature selection using stochastic gates. *In Proceedings of the 37th International Conference on Machine Learning (ICML 2020)*, 2020.

Yongliang Zhang, Joseph M Reynolds, Seon Hee Chang, Natalia Martin-Orozco, Yeonseok Chung, Roza I Nurieva, and Chen Dong. Mkp-1 is necessary for t cell activation and function. *Journal of Biological Chemistry*, 284(45):30815–30824, 2009.

