# OpenReview forum: "Self-Supervision Enhanced Feature Selection with Correlated Gates"
_ICLR.cc/2022/Conference — ICLR 2022 Spotlight_

### Official Review · Reviewer_K8kD · 2021-11-03

**Correctness:** 4
**Technical Novelty And Significance:** 3
**Empirical Novelty And Significance:** 3
**Recommendation:** 8
**Confidence:** 3

**Main Review:**

1. Equation (6) was borrowed from previous work, which makes me wonder which part of Section 4.1 is the novel contribution from this work. In addition, the last paragraph of Section 4.1 where two advantages are listed is quite hard to follow.

2. Perhaps I missed this, how does one compute the correlation structure R based on the the inputs? It is stated as `pre-specified`, is it dataset or problem dependent?

3. Regarding the experiments, most of the baselines are rather weak models. Could you elaborate among previous work which are those that also used neural networks?

4. Overall what will the model behave if none of the features are dispensable? How does it scale with the dimensionality of the data?

**Summary Of The Paper:**

This work presents some additional tweaks and tricks in using neural networks for feature selection. The main contributions include an inclusion of a parameterized masking function that selects the feature and a self-supervised component for pretraining. The overall proposal makes sense and there are strong experimental results that justifies the merits.

**Summary Of The Review:**

The experiments are rather comprehensive and illustrative. I look forward to get some feedback from the authors.

---

> ### Author Response · Authors · 2021-11-17
> **Response to Reviewer K8kD**
>
> We thank the reviewer for their feedback on our work. We have addressed all comments in the updated version of our manuscript and provide a point-by-point response below.
>
> (Section 4.1)
>
> As the reviewer pointed out, the reparameterization trick for generating multivariate Bernoulli random variables is not new. However, we would like to emphasize that our work is the first work that applies such a reparameterization trick to i) find better representations under a self-supervised learning framework and ii) discover important features under the presence of highly-correlated features.
>
> As suggested by the reviewer, we have modified the last paragraph of Section 4.1 in the updated manuscript to improve clarity as follows:
> “Using the correlation structure of the input features when generating the gate vector has the following two advantages: (i) during the Self-Supervision Phase, correlated gate vectors encourage the network not to rely on trivial signals from highly correlated features when solving the pretext tasks which may lead to inefficient pre-training and sub-optimal downstream performance; (ii) during the Supervision Phase, correlated gate vectors encourage the network to select only the most relevant features by increasing the chance that correlated features are selected jointly and thus compete against each other.”
>
>
> (Correlation Structure R)
>
> The correlation structure $R$ depends on the data and shows the inherent structure of the data with respect to how each feature is correlated with other features. We compute $R$ for each dataset following the details in Appendix C (also included in Algorithms 1 and 2).
>
> Formally, each element of correlation matrix $R$ is given as $R_{kj}=\frac{|C_{kj}|}{\sqrt{C_{kk}\cdot C_{jj}}}$ where $C_{kj} = \frac{\sum_{i=1}^{n_{u}} (x\_{k}\^{i} - \\bar{x}\_{k}\^{i})(x\_{j}\^{i} - \\bar{x}\_{j}\^{i})}{n\_{u}-1}$.
>
> (NN-based Benchmarks)
>
> Throughout all experiments, we compared SEFS to a wide range of feature selection methods and ablations of our work. While the benchmarks included several traditional methods, we compared our method to the following state-of-the-art neural network-based feature selection methods: BNNsel (Liang et al., 2018), STG (Yamada et al. 2020), and DUFS (Lindenbaum et al. 2020). In addition, we designed a variant of STG -- denoted as STG (SS) -- that extends STG to the semi-supervised setting by augmenting the loss with a reconstruction task that estimates the original features from the gated inputs.
>
>
> (No Dispensable Features)
>
> Our method can naturally avoid discarding meaningful features for the datasets with no dispensable features due to the following two aspects:
> First, we use the validation performance to determine the hyper-parameter $\beta$ that controls the number of features to be selected. If none of the features are dispensable, the hyper-parameter search will find a relatively small $\beta$, and thus the penalty term in Eq. (7) for selecting additional features will be relatively small.
> Second, minimizing the prediction loss (e.g., the cross-entropy for classification) in Eq. (7) will make the model keep the features that are important for predicting the target.
> Together with a small $\beta$ and strong signals about the target outcome from the features, our model is not likely to remove the non-dispensable features.
>
> (Scalability)
>
> Given the same number of nodes and layers in the hidden layers, the number of trainable parameters linearly increases with the number of features. Thus, from the training perspective, the computational complexity increases linearly.
>
> There are two other points that need to be considered that relate to generating the correlated gate vectors. One is that constructing the correlation matrix $R$, which is performed once at the beginning, has a computational complexity that increases quadratically with the number of the features. The other is related to generating the correlated gate vectors, which requires us to sample from a multivariate Bernoulli distribution. While this in theory also scales quadratically, we propose an "efficient implementation" that is described in Appendix B. Specifically, we use a block-wise matrix multiplication approach, where the complexity scales quadratically w.r.t. the largest block size (i.e., the number of features grouped in the largest block). Thus, if the largest block size remains the same, the complexity of generating the correlated gate vectors will only increase linearly with the feature dimension (since the number of blocks would increase linearly). In our experiments, this allowed us to scale to a dataset (PBMC) with c. 20,000 features without difficulty on a single GPU machine (compute specifications are provided in Appendix B). We have added some additional comments about the scalability of our approach in Appendix B.

---

### Official Review · Reviewer_gJvk · 2021-11-03

**Correctness:** 4
**Technical Novelty And Significance:** 3
**Empirical Novelty And Significance:** 3
**Recommendation:** 8
**Confidence:** 2

**Main Review:**

The paper is well motivated and the method is clearly presented with all its technique points delivered. Several concerns during my reading are well addressed: the Bernoulli distribution reparameterization, the lasso regularizer term's relaxation, the ablation study on SS phase, the performance gain over other baselines etc.


In section 3, the formulation around Eq1, Eq2 may have contradicting ideas on how to set values for features not selected. It reads around Eq 1, features not selected will take an out-of-distribution value "*", while in Eq 2, features not selected will take their mean values. Do authors have any comments or corrections here? Readers may be confused by the two formulations.


Another similar tabular data prediction task is CTR prediction for ads recommendation, this method may have potential to be applied there.

**Summary Of The Paper:**

This paper presents a novel feature selection method for tabular data prediction. It tackles two challenges by respective novel designs, i.e. labeled data scarcity issue by an unsupervised self-supervision phase, and feature correlation issue by multivariate Bernoulli gate vector with learnable correlation matrix. The experiments on one synthetic and two medicine/biology datasets demonstrate effectiveness.

**Summary Of The Review:**

This paper is impressive. Its technique is novel and clear. Experiment is convincing and solid. A clear accept.

---

> ### Author Response · Authors · 2021-11-17
> **Response to Reviewer gJvk**
>
> We thank the reviewer for their kind feedback on our work. We have addressed all the comments in the updated version of our manuscript and provide a point-by-point response below.
>
> (Clarification of Eq. (1) and (2))
>
> We used the notation "$\*$" to denote that a feature is not selected and not used to predict the target outcome. When implemented in neural networks, it is common practice to assign an arbitrary value (e.g. 0) in place of $\*$. In our work, we set $\*=\bar{\mathbf{x}}$ where $\\bar{\\mathbf{x}}=\\mathbb{E}\_{\\mathbf{x}\\sim p\_{X}}[\\mathbf{x}]$ (i.e. the mean). This is to enable the training of neural networks while varying the input subset and to resolve any issue when a feature having zero value (e.g., turned-off genes) has a specific meaning.
>
> We added the following footnote in the manuscript to clarify this: “To enable neural networks to be trained with varying feature subsets, we set $\*=\\bar{\\mathbf{x}}$.”
>
> (CTR Prediction)
>
> We thank the reviewer for the recommendation. Though we are not familiar with CTR prediction, we do believe that selecting important feature subsets for tabular data -- especially using both labeled and unlabeled -- is a ubiquitous problem providing a wide range of potential applications for SEFS.

---

### Official Review · Reviewer_uaxu · 2021-11-04

**Correctness:** 4
**Technical Novelty And Significance:** 3
**Empirical Novelty And Significance:** 3
**Recommendation:** 10
**Confidence:** 4

**Main Review:**

Comments:
I think the dataset descriptions could have been earlier, it’s helpful to read them prior to some of the methods descriptions to better understand the approach.

Given the focus on genetic data in the introduction it may be helpful to cite RELIEF-based approaches.

The introduction mentions WGS and the high dimensionality inherent to it but the datasets used are several orders of magnitude less features. Either the introduction should be updated to better reflect the datasets used (e.g., not full sequence data) or the potential application to genotypes (~2M features) or sequencing should be explored. Importantly, it seems this would require a streaming approach or other algorithmic changes if the data did not fit into GPU memory. In any event, It would be helpful to see the authors thoughts of about whether these methods could be combined using dimensionality reduction as pre- or post-processing in addition to, instead of as a replacement for their approach (as in Table 1).

The name - SEFS (no SS) isn't consistent from table 1. Why isn't SEFS (no SS) shown in table 1? My understanding is that it is neither the autoencoder or independent Bernoulli approach, but I was confused about why it couldn't be included in the initial experiment.

I was confused about why Table 1 does not include any of the benchmark approaches from Table S4? At first I thought the description of performance comparison in terms of both TPR and downstream predictive performance was incorrect because I did not see the supplemental table. The discussion about traditional methods performing poorly on feature 1 seems important enough to be included in the core manuscript as it provides intuition about the advantages of the author's approach.

Minor comments:
Figure 4's legend is a little confusing - lower the better, but the rank is lowest at the top of the figure. I believe this could be easily clarified.

Language should be clarified / copy-edited prior to publication
Just as a single example - Changing via to by simplifies this sentence in abstract:
First, we pre-train the network using unlabeled samples within a self-supervised learning framework via solving pretext tasks that require the network to learn informative representations from partial feature sets.

Supplement - SEFS outperforms all benchmarks - benchmarks is misspelled


**Summary Of The Paper:**

This work aims to use self-supervision to identify the most useful features for downstream tasks particularly in the context of correlated features. This is an important aspect that is lacking from many feature selection approaches commonly used within the highly correlated datasets of healthcare. This approach - using both feature vector reconstruction and gate vector estimation appears to provide better results for both true structure estimation in a  dataset with known truth as well as downstream predictive performance in real world data sets.

**Summary Of The Review:**

I found this work to be focused on an important problem as well as technically interesting and correct. I believe the authors could substantially improve the presentation of the work (e.g., condensing intro sections, reworking the order and going deeper into results - both quantitative presentation as well as the interpretation in the discussion). I believe the authors could do this during the camera-ready phase and therefore recommend this work be accepted with this feedback prior to publication.

---

> ### Author Response · Authors · 2021-11-17
> **Response to Reviewer uaxu**
>
> We thank the reviewer for their positive feedback on our work. We have addressed their comments in the updated version of our manuscript and provide a point-by-point response below.
>
> (Additional benchmark: RELIEF)
>
> We thank the reviewer for suggesting RELIEF and have added this reference to the related work. In addition, we report below the experimental results of RELIEF (implemented based on the Python package sklearn-relief) on the Two-Moons, UKCF, PBME, and CCLE datasets. Note that the performance of RELIEF is generally relatively weak compared to the other benchmarks and significantly outperformed by SEFS.
>
> The average TPRs on the Two-Moons dataset (same setup as Table S.4.).
> - ($n_{\ell}=20, n_{\ell}=1000$)  Feat 1:  0.20,  Feat 2: 0.83,  Average: 0.515
> - ($n_{\ell}=80, n_{\ell}=1000$)  Feat 1:  0.09,  Feat 2: 1.00,  Average: 0.545
>
> The AUROC and AUPRC performance for the UKCF dataset (same setup as Table 2).
> - (AUROC)  0.604 $\pm$ 0.054  (AUPRC)  0.181 $\pm$ 0.036
>
> The AUROC and AUPRC performance for the PBMC dataset (same setup as Table 3).
> - (AUROC) 0.531$\pm$ 0.015  (AUPRC)  0.525 $\pm$ 0.014
>
> The MSE performance for the CCLE dataset (same setup as Table S.5).
> - AZD6244  ($|S|=5$) 2.017 $\pm$ 0.111,   ($|S|=10$) 2.200 $\pm$ 0.103
> - Erlotinib  ($|S|=5$) 0.675 $\pm$ 0.051,   ($|S|=10$) 0.640 $\pm$ 0.048
> - Irinotecan ($|S|=5$) 1.641 $\pm$ 0.085,   ($|S|=10$) 1.754 $\pm$ 0.087
> - Laptinib ($|S|=5$) 0.767 $\pm$ 0.086,   ($|S|=10$) 0.667 $\pm$ 0.042
> - PLX4720  ($|S|=5$) 0.652 $\pm$ 0.054,   ($|S|=10$) 0.638 $\pm$ 0.085
> - Topotecan ($|S|=5$) 1.823 $\pm$ 0.053,   ($|S|=10$) 1.976 $\pm$ 0.087
>
> (WGS)
>
> In our introduction, we primarily mentioned next-generation sequencing (NGS) as it is used to determine entire RNA sequences (for example, we used a dataset with approximately 20,000 gene expressions in our experiments). We apologize if any confusion was caused by mentioning NGS in this context since it was not our intention to refer to whole-genome sequencing (WGS) specifically. We have updated our introduction to make this clearer.
>
> (Potential application to genome-wide association)
>
> When considering a very high-dimensional feature dimension (such as the whole-genome sequence) various engineering solutions exist, such as parallelizing GPU computation as seen when training incredibly large networks in other domains. However, as noted by the reviewer, we can significantly reduce the computational complexity of feature selection using SEFS by performing a pre-selection of features if necessary. More specifically, we can perform a GWAS-like two-sample test but with a significantly less strict threshold and without LD pruning since our method can benefit from the correlation structure in the data.
>
> (Clarification of Table 1.)
>
> Please allow us to clarify Table 1, which contains slightly different experimental results to Figure 3 and Table S.4. The results in Table 1 highlight the importance of using our proposed self-supervised learning -- i.e., using multi-Bernoulli distribution for generating the correlated gate vectors (denoted as SEFS) -- over conventional approaches -- i.e., using a conventional autoencoder (denoted as SEFS (AE)) and using independent Bernoulli distribution for generating the gate vector (denoted as SEFS (indep)). More specifically, we show that applying multi-Bernoulli distribution to pre-train the encoder in the self-supervision phase is important to find "meaningful" representations, which is evidenced by having higher AUROC/AURPC performance compared to the two aforementioned benchmarks when an MLP is trained based on the learned representations given only the ground-truth features (i.e., setting $m_1 = m_2 = 1$ and $m_k = 0$ for $k \notin \{1, 2\} $). Since Table 1 is comparing the performance of the learned representations rather than a complete feature selection algorithm, it is not possible to include SEFS (no SS), since this ablation does not involve a self-supervision phase.
> To clarify this, we have changed the labels in Table 1. Furthermore, as per the reviewer’s suggestion, we have also updated the manuscript to include the discussion about traditional methods performing poorly on feature 1 (i.e. $x_{1}$) in the core manuscript.
>
> (Figure 4 Legend)
>
> We thank the reviewer and apologize for any confusion caused. We have revised the caption accordingly.
>
> (Other Minor Comments)
>
> We thank the reviewer and apologize for the typos which have been corrected in the updated manuscript

---

> > ### Comment · Reviewer_uaxu · 2021-11-19
> > **response to review**
> >
> > I found the response to be thorough and appropriate. I've adjusted my score as such.

---

### Decision · Program_Chairs · 2022-01-20

**Decision:**

Accept (Spotlight)

**Comment:**

This paper proposes a feature selection method to identify features for downstream supervised tasks, focused on addressing challenges with sample scarcity and feature correlations.  The proposed approach is highly motivating in biological and medical applications.  Reviewers pointed out various strengths including potential high impacts in biomedical applications, technical novelty and significance, and comprehensive and illustrative experiments.  The authors adequately addressed major concerns raised by reviewers.